# Online Control in Population Dynamics

**Noah Golowich**[*]    **Elad Hazan**[†]    **Zhou Lu**[‡]    **Dhruv Rohatgi**[§]    **Y. Jennifer Sun**[¶]

## Abstract

The study of population dynamics originated with early sociological works but has since extended into many fields, including biology, epidemiology, evolutionary game theory, and economics. Most studies on population dynamics focus on the problem of *prediction* rather than *control*. Existing mathematical models for population control are often restricted to specific, noise-free dynamics, while real-world population changes can be complex and adversarial.

To address this gap, we propose a new framework based on the paradigm of online control. We first characterize a set of linear dynamical systems that can naturally model evolving populations. We then give an efficient gradient-based controller for these systems, with near-optimal regret bounds with respect to a broad class of linear policies. Our empirical evaluations demonstrate the effectiveness of the proposed algorithm for population control even in non-linear models such as SIR and replicator dynamics.

## 1 Introduction

Dynamical systems involving populations are ubiquitous in describing processes that arise in natural environments. As one example, the SIR model [26] is a fundamental concept in epidemiology, used to describe the spread of infectious diseases within a population. It divides the population into three groups – Susceptible (S), Infected (I) and Removed (R). A *susceptible* individual has not contracted the disease but has the chance to be infected if interacting with an *infected* individual. A *removed* individual either has recovered from the disease and gained immunity or is deceased. The population evolves over time according to three ordinary differential equations:

$$\frac{dS}{dt} = -\beta IS \ , \ \frac{dI}{dt} = \beta IS - \theta I \ , \ \frac{dR}{dt} = \theta I, \tag{1}$$

for constants $\beta, \theta > 0$ representing the infection and recovery rate, respectively. Numerous extensions of this basic model have been proposed to better capture how epidemics evolve and spread [8].

Beyond epidemiology, population dynamics naturally arise in many other fields, notably evolutionary game theory [23], biology [14, 7], and the analysis of genetic algorithms [39, 25]. For any dynamical system, it is natural to ask how it might best be *controlled*. For instance, controlling the spread of an infectious disease while minimizing externalities is a problem of significant societal importance.

In many natural models, these dynamics tend to be nonlinear, so the control problem is often computationally intractable. Existing algorithms are designed on a case-by-case basis by positing a specific system of differential equations, and then numerically or analytically solving for the optimal controller. Unfortunately, this approach is not robust to adversarial shocks to the system and cannot adapt to time-varying cost functions.

---

[*]MIT. `nzg@mit.edu`.

[†]Google DeepMind & Princeton University. `ehazan@princeton.edu`.

[‡]Princeton University. `zhoul@princeton.edu`.

[§]MIT. `drohatgi@mit.edu`.

[¶]Princeton University. `ys7849@princeton.edu`.

38th Conference on Neural Information Processing Systems (NeurIPS 2024).

**A new approach for population control.** In this paper we propose a generic and robust methodology for population control, drawing on the framework and tools from online non-stochastic control theory to obtain a computationally efficient gradient-based method of control. In online non-stochastic control, at every time $t = 1, \ldots, T$, the learner is faced with a state $x_t$ and must choose a control $u_t$. The learner then incurs cost according to some time-varying cost function $c_t(x_t, u_t)$ evaluated at the current state/control pair, and the state evolves as:

$$x_{t+1} := f(x_t, u_t) + w_t, \tag{2}$$

where $f$ describes the (known) discrete-time dynamics, $x_{t+1}$ is the next state, and $w_t$ is an adversarially-chosen perturbation. A *policy* is a mapping from states to controls. The goal of the learner is to minimize *regret* with respect to some rich policy class $\Pi$, formally defined by

$$\mathbf{regret}_\Pi = \sum_{t=1}^{T} c_t(x_t, u_t) - \min_{\pi \in \Pi} \sum_{t=1}^{T} c_t(x_t^\pi, u_t^\pi), \tag{3}$$

where $(x_t^\pi, u_t^\pi)$ is the state/control pair at time $t$ had policy $\pi$ been carried out since time 1.

As with prior work in online control [1], our method is theoretically grounded by regret guarantees for a broad class of Linear Dynamical Systems (LDSs). The key algorithmic and technical challenge we overcome is that **prior methods only give regret bounds against comparator policies that *strongly stabilize* the LDS** (Definition 4). Such policies force the magnitude of the state to decrease exponentially fast in the absence of noise. Unfortunately, for applications to population dynamics, even the assumption that such policies *exist* – let alone perform well – is fundamentally unreasonable, since it essentially implies that the population can be made to exponentially shrink.

A priori, one might hope to generically overcome this issue, by broadening the comparator class to all policies that *marginally stabilize* the LDS (informally, these are policies under which the magnitude of the state does not blow up). But we show that, in general, it is impossible to achieve sub-linear regret against that class – a result that may be of independent interest in online control:[6]

**Theorem 1** (Informal statement of Theorem 25)**.** *There is a distribution $\mathcal{D}$ over LDSs with state space and control space given by $\mathbb{R}$, such that any online control algorithm on a system $\mathcal{L} \sim \mathcal{D}$ incurs expected regret $\Omega(T)$ against the class of time-invariant linear policies that marginally stabilize $\mathcal{L}$.*

For general LDSs, it's not obvious if there is a natural "intermediate" comparator class that does not require strong stabilizability and does enable control with low regret. However, systems that model *populations* possess rich additional structure, since they can be interpreted as controlled Markov chains.[7] In this paper, leveraging that structure, we design an algorithm `GPC-Simplex` for online control that applies to LDSs constrained to the *simplex* (Definition 3), and achieves strong regret bounds against a natural comparator class of policies with bounded *mixing time* (Definition 6).

## 1.1 Our Results

Throughout this work, we model a *population* as a distribution over $d$ different categories, evolving over $T$ discrete timesteps. For simplicity, we assume that $u_t$ is a $d$-dimensional real vector.

**Theoretical guarantees for online population control.** We introduce the *simplex LDS* model (Definition 3), which is a modification of the standard LDS model (Definition 9) that ensures the states $(x_t)_t$ always represent valid distributions, i.e. never leave the simplex $\Delta^d$. Informally, given state $x_t \in \Delta^d$ and control $u_t \in \mathbb{R}^d_{\geqslant 0}$ with $\|u_t\|_1 \leqslant 1$, the next state is

$$x_{t+1} = (1 - \gamma_t) \cdot [(1 - \|u_t\|_1)Ax_t + Bu_t] + \gamma_t \cdot w_t,$$

where $A, B$ are known stochastic matrices, $\gamma_t \in [0, 1]$ is the observed *perturbation strength*,[8] and $w_t \in \Delta^d$ is an unknown perturbation. The perturbation $w_t$ can be interpreted as representing an

---

[6][22, 20] showed that the *prediction* task in a marginally stable LDS can be solved with sublinear regret via spectral filtering if the state transition matrix is symmetric. The construction for Theorem 1 has symmetric transition matrices, so the result in a sense separates analogous prediction and control tasks.

[7]Markov decision processes can also be thought of as controlled Markov chains. However, in that setting the controls/actions are at the individual level, whereas we are concerned with controls at the population level, as motivated by applications to epidemiology, evolutionary game theory, and other fields.

[8]See Appendix B for discussion about this modelling assumption.

adversary that can add individuals from a population with distribution $w_t$ to the population under study. Intuitively, $u_t$ represents a distribution over $d$ possible interventions as well as a "null intervention".

For any simplex LDS $\mathcal{L}$ and mixing time parameter $\tau > 0$, we define a class $\mathcal{K}_\tau^\triangle(\mathcal{L})$ (Definition 6), which roughly consists of the linear time-invariant policies under which the state of the system would mix to stationarity in time $\tau$, in the absence of noise. Our main theoretical contribution is an algorithm GPC-Simplex that achieves low regret against this policy class:

**Theorem 2** (Informal version of Theorem 7). *Let $\mathcal{L}$ be a simplex LDS on $\Delta^d$, and let $\tau > 0$. For any adversarially-chosen perturbations $(w_t)_t$, perturbation strengths $(\gamma_t)_t$, and convex and Lipschitz cost functions $(c_t)_t$, the algorithm* GPC-Simplex *performs $T$ steps of online control on $\mathcal{L}$ with regret $\tilde{O}(\tau^{7/2}\sqrt{dT})$ against $\mathcal{K}_\tau^\triangle(\mathcal{L})$.*

Finally, analogously Theorem 1, we show that the mixing time assumption cannot be removed: it is impossible to achieve sub-linear regret (for online control of a simplex LDS) against the class of all linear time-invariant policies (Theorem 8).

**Experimental evaluations.** To illustrate the practicality of our results, we apply (a generalization of) GPC-Simplex to controlled versions of (a) the SIR model for disease transmission (Section 4), and (b) the replicator dynamics from evolutionary game theory (Appendix H). In the former, closed-form optimal controllers are known in the absence of perturbations [27]. We find that GPC-Simplex *learns* characteristics of the optimal control (e.g. the "turning point" phase transition where interventions stop once herd immunity is reached). Moreover, our algorithm is robust even in the presence of adversarial perturbations, where previous theoretical results no longer apply. In the latter, we demonstrate that even when the control affects the population only indirectly, through the replicator dynamics *payoff matrix*, GPC-Simplex can learn to control the population effectively, and is more robust to noisy cost functions than a one-step best response controller.

## 1.2 Related work

**Online non-stochastic control.** In recent years, the machine learning community has witnessed an increasing interest in non-stochastic control problems (e.g. [1, 38, 19]). Unlike the classical setting of stochastic control, in non-stochastic control the dynamics are subject to time-varying, adversarially chosen perturbations and cost functions. See [21] for a survey of prior results. Most relevant to our work is the Gradient Perturbation Controller (GPC) for controlling general LDSs [1]. All existing controllers only provide provable regret guarantees against policies that strongly stabilize the system.

**Population growth models.** There is extensive research on modeling the evolution of populations in sociology, biology and economics. Besides the pioneering work of [33], notable models include the SIR model from epidemiology [26], the Lotka–Volterra model for predator-prey dynamics [32, 40] and the replicator dynamics from evolutionary game theory [23]. Recent years have seen intensive study of controlled versions of the SIR model – see e.g. empirical work [10], vaccination control models [13], and many others [15, 12, 30, 17]. Most relevant to our work is the *quarantine control model*, where the control reduces the effective transmission rate. Some works consider optimal control in the noiseless setting [27, 5]; follow-up work [34] considers a budget constraint on the control. None of these prior works can handle the general case of adversarial noise and cost functions.

## 2 Definitions and setup

**Notation.** Denote $\mathbb{S}^d := \left\{ M \in [0,1]^{d \times d} \ : \ \sum_{i=1}^d M_{i,j} = 1 \ \forall j \in [d] \right\}$ as the set of $d \times d$ column-stochastic matrices. For $a > 0$, define $\mathbb{S}_a^d := \{a \cdot M \ : \ M \in \mathbb{S}^d\}$ and $\mathbb{S}_{\leqslant a}^d := \bigcup_{0 \leqslant a' \leqslant a} \mathbb{S}_{a'}^d$. Let $\Delta^d$ denote the simplex in $\mathbb{R}^d$. Similarly, we define $\Delta_\alpha^d := \alpha \cdot \Delta^d$ and $\Delta_{\leqslant \alpha}^d := \bigcup_{0 \leqslant \alpha' \leqslant \alpha} \Delta_{\alpha'}^d$. Given a square matrix $M \in \mathbb{R}^{d \times d}$, let $M_{\cdot,j}$ denote the $j$th column of $M$. We consider the following matrix norms: $\|M\|$ denotes the spectral norm of $M$, $\|M\|_{2,1}^2 := \sum_{j=1}^d \|M_{\cdot,j}\|_1^2$ is the sum of the squares of the $\ell_1$ norms of the columns of $M$, and $\|M\|_{1 \to 1} := \sup_{x \in \mathbb{R}^d : \|x\|_1 = 1} \|Mx\|_1$.

## 2.1 Dynamical systems

The standard model in online control is the *linear dynamical system (LDS)*. We define a *simplex LDS* to be an LDS where the state of the system always lies in the simplex. This requires enforcing certain constraints on the transition matrices, the control, and the noise:

**Definition 3** (Simplex LDS). Let $d \in \mathbb{N}$. A *simplex LDS on* $\Delta^d$ is a tuple

$$\mathcal{L} = (A, B, \mathcal{I}, x_1, (\gamma_t)_{t \in \mathbb{N}}, (w_t)_{t \in \mathbb{N}}, (c_t)_{t \in \mathbb{N}}),$$

where $A, B \in \mathbb{S}^d$ are the *transition matrices*; $\mathcal{I} \subseteq \Delta^d_{\leqslant 1}$ is the *valid control set*; $x_1 \in \Delta^d$ is the *initial state*; $\gamma_t \in [0, 1]$, $w_t \in \Delta^d$ are the *noise strength* and *noise value* at time $t$; and $c_t : \Delta^d \times \mathcal{I} \to \mathbb{R}$ is the *cost function* at time $t$. These parameters define a dynamical system where the state at time $t = 1$ is $x_1$. For each $t \geqslant 1$, given state $x_t$ and control $u_t \in \mathcal{I}$ at time $t$, the state at time $t + 1$ is

$$x_{t+1} = (1 - \gamma_t) \cdot [(1 - \|u_t\|_1) A x_t + B u_t] + \gamma_t \cdot w_t, \tag{4}$$

and the cost incurred at time $t$ is $c_t(x_t, u_t)$.

Note that since the set of possible controls $\mathcal{I}$ is contained in $\Delta^d_{\leqslant 1}$, the states $(x_t)_t$ are guaranteed to remain within the simplex for all $t$. In this paper, we will assume that $\mathcal{I} = \bigcup_{\alpha \in [\underline{\alpha}, \overline{\alpha}]} \Delta^d_\alpha$, for some parameters $\underline{\alpha}, \overline{\alpha} \in [0, 1]$, which represent lower and upper bounds on the strength of the control.

**Online non-stochastic control.** Let $\mathcal{L} = (A, B, x_1, (\gamma_t)_{t \in \mathbb{N}}, (w_t)_{t \in \mathbb{N}}, (c_t)_{t \in \mathbb{N}}, \mathcal{I})$ be a simplex LDS and let $T \in \mathbb{N}^+$. We assume that the transition matrices $A, B$ are known to the controller at the beginning of time, but the perturbations $(w_t)_{t=1}^T$ are unknown. At each step $1 \leqslant t \leqslant T$, the controller observes $x_t$ and $\gamma_t$, plays a control $u_t \in \mathcal{I}$, and then observes the cost function $c_t$ and incurs cost $c_t(x_t, u_t)$. The system then evolves according to Eq. (4). Note that our assumption that the controller observes $\gamma_t$ contrasts with some of the existing work on nonstochastic control [1, 21], in which no information about the adversarial disturbances is known. In Appendix B, we justify the learner's ability to observe $\gamma_t$ by observing that in many situations, the learner observes the *counts* of individuals in a populations (in addition to their *proportions*, represented by the state $x_t$), and that this additional information allows computation of $\gamma_t$.

The goal of the controller is to minimize regret with respect to some class $\mathcal{K} = \mathcal{K}(\mathcal{L})$ of comparator policies. Formally, for any fixed dynamical system and any time-invariant and Markovian policy $K : \Delta^d \to \mathcal{I}$, let $(x_t(K))_t$ and $(u_t(K))_t$ denote the counterfactual sequences of states and controls that would have been obtained by following policy $K$. Then the regret of the controller on observed sequences $(x_t)_t$ and $(u_t)_t$ with respect to $\mathcal{K}$ is

$$\mathbf{regret}_\mathcal{K} := \sum_{t=1}^T c_t(x_t, u_t) - \inf_{K \in \mathcal{K}} \sum_{t=1}^T c_t(x_t(K), u_t(K)).$$

The following assumption on the cost functions of $\mathcal{L}$ is standard in online control [1, 3, 38, 37]:

**Assumption 1.** *The cost functions* $c_t : \Delta^d \times \mathcal{I} \to \mathbb{R}$ *are convex and $L$-Lipschitz, in the following sense: for all $x, x' \in \Delta^d$ and $u, u' \in \mathcal{I}$, we have $|c_t(x, u) - c_t(x', u')| \leqslant L \cdot (\|x - x'\|_1 + \|u - u'\|_1)$.*

## 2.2 Comparator class and spectral conditions

In prior works on non-stochastic control for linear dynamical systems [1], the comparator class $\mathcal{K} = \mathcal{K}_{\kappa,\rho}(\mathcal{L})$ is defined to be the set of linear, time-invariant policies $x \mapsto Kx$ where $K \in \mathbb{R}^{d \times d}$ $(\kappa, \rho)$-*strongly stabilizes* $\mathcal{L}$:

**Definition 4.** A matrix $M \in \mathbb{R}^{d \times d}$ is $(\kappa, \rho)$-*strongly stable* if there is a matrix $H \in \mathbb{R}^{d \times d}$ so that $\|H^{-1}MH\| \leqslant 1 - \rho$ and $\|M\|, \|H\|, \|H^{-1}\| \leqslant \kappa$. A matrix $K \in \mathbb{R}^{d \times d}$ is said to $(\kappa, \rho)$-*strongly stabilize* an LDS with transition matrices $A, B \in \mathbb{R}^{d \times d}$ if $A + BK$ is $(\kappa, \rho)$-strongly stable.

The regret bounds against $\mathcal{K}_{\kappa,\rho}(\mathcal{L})$ scale with $\rho^{-1}$, and so are vacuous for $\rho = 0$ [1]. Unfortunately, in the simplex LDS setting, no policies satisfy the analogous notion of strong stability (see discussion in Section 3) unless $\rho = 0$. Intuitively, the reason is that a $(\kappa, \rho)$-strongly stable policy with $\rho > 0$ makes the state converge to 0 in the absence of noise.

What is a richer but still-tractable comparator class for a simplex LDS? We propose the class of linear, time-invariant policies under which (a) the state of the LDS *mixes*, when viewed as a distribution, and (b) the level of control $\|u_t\|_1$ is independent of the state $x_t$. Formally, we make the following definitions:

**Definition 5.** Given $t \in \mathbb{N}$ and a matrix $X \in \mathbb{S}^d$ with unique stationary distribution $\pi \in \Delta^d$, we define $D_X(t) := \sup_{p \in \Delta^d} \|X^t p - \pi\|_1$ and $\bar{D}_X(t) := \sup_{p,q \in \Delta^d} \|X^t \cdot (p - q)\|_1$. Moreover we define $t^{\mathsf{mix}}(X, \varepsilon) := \min_{t \in \mathbb{N}}\{t \ : \ D_X(t) \leqslant \varepsilon\}$ for each $\epsilon > 0$, and we write $t^{\mathsf{mix}}(X) := t^{\mathsf{mix}}(X, 1/4)$.[9]

**Definition 6** (Mixing a simplex LDS). Let $\mathcal{L}$ be a simplex LDS with transition matrices $A, B \in \mathbb{S}^d$ and control set $\mathcal{I} = \bigcup_{\alpha \in [\underline{\alpha}, \overline{\alpha}]} \Delta^d_\alpha$. A matrix $K \in \mathbb{S}^d_{[\underline{\alpha}, \overline{\alpha}]}$ is said to $\tau$-*mix* $\mathcal{L}$ if $t^{\mathsf{mix}}(\mathbb{A}_K) \leqslant \tau$, where

$$\mathbb{A}_K := (1 - \|K\|_{1 \to 1}) \cdot A + BK \in \mathbb{S}^d. \tag{5}$$

We define the comparator class $\mathcal{K}^\triangle_\tau = \mathcal{K}^\triangle_\tau(\mathcal{L})$ as the set of linear, time-invariant policies $x \mapsto Kx$ where $K \in \mathbb{S}^d_{[\underline{\alpha}, \overline{\alpha}]}$ $\tau$-mixes $\mathcal{L}$.

Notice that for any $K \in \mathbb{S}^d_{[\underline{\alpha}, \overline{\alpha}]}$, the linear policy $u_t := Kx_t$ always plays controls in the control set $\mathcal{I}$, and the dynamics Eq. (4) under this policy can be written as $x_{t+1} = (1 - \gamma_t) \cdot \mathbb{A}_K x_t + \gamma_t \cdot w_t$.

Notice that by considering the comparator class $\mathcal{K}^\triangle_\tau$, we require the control norm to be independent of the state. This assumption is needed for technical reasons: without it, since $Ax$ is multiplied by $1 - \|u\|_1$ in the transition dynamics (see Equation (4)), even a "linear" policy $u := Kx$ does not induce a linear transition. Hence, it would no longer be clear how one might define mixing time of a linear policy. It is a very interesting question whether there is a more natural (yet still tractable) definition of a simplex LDS that avoids this issue.

# 3 Online Algorithm and Theoretical Guarantee

In this section, we describe our main upper bound and accompanying algorithm for the setting of online control in a simplex LDS $\mathcal{L}$. As discussed above, we assume that the set of valid controls is given by $\mathcal{I} = \bigcup_{\alpha \in [\underline{\alpha}, \overline{\alpha}]} \Delta^d_\alpha$, for some constants $0 \leqslant \underline{\alpha} \leqslant \overline{\alpha} \leqslant 1$, representing lower and upper bounds on the strength of the control.[10]

For convenience, we write $\alpha_t := \|u_t\|_1$ and $u'_t = u_t / \alpha_t \in \Delta^d$ (if $\alpha_t = 0$, we set $u'_t := 0$). The dynamical system Eq. (4) can then be expressed as follows:

$$x_{t+1} = (1 - \gamma_t) \cdot ((1 - \alpha_t) \cdot Ax_t + \alpha_t \cdot Bu'_t) + \gamma_t \cdot w_t. \tag{6}$$

We aim to obtain a regret guarantee as in Eq. (3) with respect to some rich class of *comparator policies* $\mathcal{K}^\triangle$. As is typical in existing work on linear nonstochastic control, we take $\mathcal{K}^\triangle$ to be a class of time-invariant *linear* policies, i.e. policies that choose control $u_t := Kx_t$ at time $t$ for some matrix $K \in \mathbb{R}^{d \times d}$. In the standard setting of nonstochastic control, it is typically further assumed that all policies in the comparator class strongly stabilize the LDS (Definition 4).[11] The naive generalization of such a requirement in our setting would be that $\mathbb{A}_K$ is strongly stable; however, this is impossible, since no stochastic matrix can be strongly stable. Instead, we aim to compete against the class $\mathcal{K}^\triangle = \mathcal{K}^\triangle_\tau(\mathcal{L})$ of time-invariant linear policies that (a) have fixed level of control in $[\underline{\alpha}, \overline{\alpha}]$, and (b) $\tau$-mix $\mathcal{L}$ (Definition 6). We view the second condition as a natural distributional analogue of strong stabilizability; the first condition is needed for $\tau$-mixing to even be well-defined.

**Algorithm description.** Our main algorithm, `GPC-Simplex` (Algorithm 1), is a modification of the `GPC` algorithm [1, 21]. As a refresher, `GPC` chooses the controls $u_t$ by learning a *disturbance-action policy*: a policy $u_t := \bar{K}x_t + \sum_{i=1}^H M^{[i]} w_{t-i}$, where $\bar{K}$ is a known, fixed matrix that strongly stabilizes the LDS; $w_{t-1}, \ldots, w_{t-H}$ are the recent noise terms; and $M^{[1]}, \ldots, M^{[H]}$ are learnable, matrix-valued parameters which we abbreviate as $M^{[1:H]}$. The key advantage of this parametrization

---

[9]If $X$ does not have a unique stationary distribution, we say that all of these quantities are infinite.

[10]While our techniques allow some more general choices for $\mathcal{I}$, we leave a full investigation of general $\mathcal{I}$ for future work.

[11]Such an assumption cannot be dropped in light of Theorem 1.

of policies (as opposed to a simpler parametrization such as $u_t = Kx_t$ for a parameter $K$) is that the entire trajectory is *linear* in the parameters, and not a high-degree polynomial. Thus, optimizing the cost of a trajectory over the class of disturbance-action policies is a convex problem in $M^{[1:H]}$.

But why is the class of disturbance-action policies expressive enough to compete against the comparator class? This is where GPC crucially uses strong stabilizability. Notice that in the absence of noise, every disturbance-action policy is identical to the fixed policy $u_t := \bar{K}x_t$. This is fine when $\bar{K}$ and the comparator class are strongly stabilizing, since in the absence of noise, *all* strongly stabilizing policies rapidly force the state to 0, and thus incur very similar costs in the long run. But in the simplex LDS setting, strong stabilizability is impossible. While all policies in $\mathcal{K}^{\triangle}$ mix the LDS, they may mix to different states, which may incur different costs. There is no reason to expect that an arbitrary $\bar{K} \in \mathcal{K}^{\triangle}$, chosen before observing the cost functions, will have low regret against all policies in $\mathcal{K}^{\triangle}$.

We fix this issue by enriching the class of disturbance-action policies with an additional parameter $p \in \Delta^d$ which, roughly speaking, represents the desired stationary distribution to which $x_t$ would converge, in the absence of noise, as $t \to \infty$. It is unreasonable to expect prior knowledge of the optimal choice of $p$, which depends on the not-yet-observed cost functions. Thus, GPC-Simplex instead *learns* $p$ together with $M^{[1:H]}$. We retain the property that the requisite online learning problem is convex in the parameters, and therefore can be efficiently solved via an online convex optimization algorithm (as discussed in Appendix C.1, we use lazy mirror descent, LazyMD). One advantage of GPC-Simplex over GPC is that the former requires no knowledge of the fixed "reference" policy $\bar{K}$ (which, in the context of GPC, had to be strongly stabilizing). While such $\bar{K}$ is needed in the context of GPC to bound a certain approximation error involving the cost functions, in the context of GPC-Simplex this approximation error may be bounded by some simple casework involving properties of stochastic matrices (see Appendix C.3).

Formally, for parameters $a_0 \in [\underline{\alpha}, \overline{\alpha}]$ and $H \in \mathbb{N}$, GPC-Simplex considers a class of policies parametrized by the set $\mathcal{X}_{d,H,a_0,\overline{\alpha}} := \bigcup_{a \in [a_0,\overline{\alpha}]} \Delta_a^d \times (\mathbb{S}_a^d)^H$. We abbreviate elements $(p, (M^{[1]}, \ldots, M^{[H]})) \in \mathcal{X}_{d,H,a_0,\overline{\alpha}}$ by $(p, M^{[1:H]})$. The high level idea of GPC-Simplex, like that of GPC, is to perform online convex optimization on the domain $\mathcal{X}_{d,H,a_0,\overline{\alpha}}$ (Line 10). At each time $t$, the current iterate $(p_t, M_t^{[1:H]})$, which defines a policy $\pi^{p_t, M_t^{[1:H]}}$, is used to choose the control $u_t$. The optimization subroutine then receives a new loss function $\ell_t : \mathcal{X}_{d,H,a_0,\overline{\alpha}} \to \mathbb{R}$ based on the newly observed cost function $c_t$. As with GPC, showing that this algorithm works requires showing that the policy class is sufficiently expressive. Unlike for GPC, our comparator policies are not strongly stabilizing, so new ideas are required for the proof.

We next formally define the policy $\pi^{p, M^{[1:H]}}$ associated with parameters $(p, M^{[1:H]})$, and the loss function $\ell_t$ used to update the optimization algorithm at time $t$.

**Parametrization of policies.** First, for $t \in [T]$ and $i \in \mathbb{N}^+$, we define the weights

$$\lambda_{t,i} := \gamma_{t-i} \cdot \prod_{j=1}^{i-1}(1 - \gamma_{t-j}), \qquad \bar{\lambda}_{t,i} := \prod_{j=1}^{i}(1 - \gamma_{t-j}), \qquad \lambda_{t,0} := 1 - \sum_{i=1}^{H}\lambda_{t,i}. \qquad (7)$$

We write $w_0 := x_1$, $\gamma_0 = 1$, and $w_t = 0$ for $t < 0$ as a matter of convention.[12] $\lambda_{t,i}$ can be interpreted as the "influence of perturbation $w_{t-i}$ on the state $x_t$", and $\bar{\lambda}_{t,i}$ can be interpreted as the "influence of perturbations prior to time step $t - i$ on the state $x_t$". An element $(p, M^{[1:H]}) \in \mathcal{X}_{d,H,a_0,\overline{\alpha}}$ induces a policy[13] at time $t$, denoted $\pi_t^{p, M^{[1:H]}}$, via the following variant of the *disturbance-action control* [1]:

$$\pi_t^{p, M^{[1:H]}}(\delta_{t-1:t-H}) := \lambda_{t,0} \cdot p + \sum_{j=1}^{H} \lambda_{t,j} \cdot M^{[j]} \delta_{t-j}. \qquad (8)$$

In Line 6 of GPC-Simplex, the control $u_t$ is chosen to be $\pi_t^{p_t, M_t^{[1:H]}}(w_{t-1:t-H})$, which belongs to $\Delta_{\|p_t\|_1}^d$ (using $\sum_{i=0}^{H} \lambda_{t,i} = 1$) and hence to the constraint set $\mathcal{I}$ (since $\|p_t\|_1 \in [a_0, \overline{\alpha}] \subset [\underline{\alpha}, \overline{\alpha}]$).

---

[12]As a result of this convention, we have $\sum_{i=1}^{t} \lambda_{t,i} = 1$ for all $t \in [T]$, and $\lambda_{t,i} = 0$ for all $i > t$.

[13]Technically, we are slightly abusing terminology here, since $\pi_t^{p, M^{[1:H]}}$ takes as input a set of the previous $H$ disturbances, $\delta_{t-1:t-H}$, as opposed to the current state $x_t$.

---

**Algorithm 1** `GPC-Simplex`: GPC for Simplex LDS

---

**Require:** Linear system $A, B$, mixing time $\tau > 0$ for comparator class, horizon parameter $H \in \mathbb{N}$, set of valid controls $\mathcal{I} = \bigcup_{\alpha \in [\underline{\alpha}, \overline{\alpha}]} \alpha \cdot \Delta^d$, total number of time steps $T$.

1: Write $\tau_A := t^{\mathsf{mix}}(A)$, and define $a_0 := \max\{\underline{\alpha}, \min\{\overline{\alpha}, \mathbb{1}\{\tau_A > 4\tau\}/(96\tau)\}\}$.

2: Initialize an instance `LazyMD` of mirror descent (Algorithm 2) for the domain $\mathcal{X}_{d, H, a_0, \overline{\alpha}}$ with the regularizer $R_{d, H}$ (defined in Appendix C.1) and step size $\eta = c\sqrt{dH \ln(d)}/(L\tau^2 \log^2(T)\sqrt{T})$, for a sufficiently small constant $c$.

3: Initialize $(p_1, M_1^{[1:H]}) \leftarrow \arg\min_{(p, M^{[1:H]}) \in \mathcal{X}_{d, H, a_0, \overline{\alpha}}} R_{d, H}(p, M^{[1:H]})$.

4: Observe initial state $x_1 \in \Delta^d$.

5: **for** $1 \leqslant t \leqslant T$ **do**

6:      Choose control $u_t := \lambda_{t,0} \cdot p_t + \sum_{i=1}^{H} M_t^{[i]} \cdot \lambda_{t,i} \cdot w_{t-i}$.

7:      Receive cost $c_t(x_t, u_t)$.

8:      Observe $x_{t+1}, \gamma_t$ and compute $w_t = \gamma_t^{-1}(x_{t+1} - (1 - \gamma_t)[(1 - \|u_t\|_1)Ax_t + Bu_t])$.   (If $\gamma_t = 0$, then set $w_t = 0$.)

9:      Define loss function $\ell_t(p, M^{[1:H]}) := c_t(x_t(p, M^{[1:H]}), u_t(p, M^{[1:H]}))$.

10:     Update $(p_{t+1}, M_{t+1}^{[1:H]}) \leftarrow \texttt{LazyMD}_t(\ell_t; (p_t, M_t^{[1:H]}))$.

---

**Loss functions.** For $(p, M^{[1:H]}) \in \mathcal{X}_{d, H, a_0, \overline{\alpha}}$, we let $x_t(p, M^{[1:H]})$ and $u_t(p, M^{[1:H]})$ denote the state and control at step $t$ obtained by following the policy $\pi_s^{p, M^{[1:H]}}$ at all time steps $s$ prior to $t$ (see Eqs. (20) and (21) in the appendix for precise definitions). We then define $\ell_t(p, M^{[1:H]})$ to be the evaluation of the adversary's cost function $c_t$ on the state-action pair $(x_t(p, M^{[1:H]}), u_t(p, M^{[1:H]}))$ (Line 9).

**Main guarantee and proof overview.** Theorem 7 gives our regret upper bound for `GPC-Simplex`:

**Theorem 7.** *Let $d, T \in \mathbb{N}$ and $\tau > 0$. Let $\mathcal{L} = (A, B, \mathcal{I}, x_1, (\gamma_t)_{t \in \mathbb{N}}, (w_t)_{t \in \mathbb{N}}, (c_t)_{t \in \mathbb{N}})$ be a simplex LDS with cost functions $(c_t)_t$ satisfying Assumption 1 for some $L > 0$. Set $H := \tau \lceil \log(2LT^3) \rceil$. Then the iterates $(x_t, u_t)_{t=1}^{T}$ of `GPC-Simplex` (Algorithm 1) with input $(A, B, \tau, H, \mathcal{I}, T)$ satisfy:*

$$\mathbf{regret}_{\mathcal{K}_\tau^\triangle(\mathcal{L})} := \sum_{t=1}^{T} c_t(x_t, u_t) - \inf_{K \in \mathcal{K}_\tau^\triangle(\mathcal{L})} \sum_{t=1}^{T} c_t(x_t(K), u_t(K)) \leqslant \tilde{O}(L\tau^{7/2} d^{1/2}\sqrt{T}),$$

*where $\tilde{O}(\cdot)$ hides only universal constants and poly-logarithmic dependence in $T$. Moreover, the time complexity of `GPC-Simplex` is $\mathrm{poly}(d, T)$.*

While for simplicity we have stated our results for obliviously chosen $(\gamma_t)_t, (w_t)_t, (c_t)_t$, since `GPC-Simplex` is deterministic the result also holds when these parameters are chosen adaptively by an adversary. See Appendix C for the formal proof of Theorem 7.

**Lower bound.** We also show that the mixing assumption on the comparator class $\mathcal{K}_\tau^\triangle(\mathcal{L})$ (Definition 6) cannot be removed. In particular, without that assumption, if the valid control set $\mathcal{I}$ is restricted to controls $u_t$ of norm at mostly roughly $O(1/T)$, then linear regret is unavoidable.[14]

**Theorem 8** (Informal statement of Theorem 30). *Let $\beta > 0$ be a sufficiently large constant. For any $T \in \mathbb{N}$, there is a distribution $\mathcal{D}$ over simplex LDSs with state space $\Delta^2$ and control space $\bigcup_{\alpha \in [0, \beta/T]} \Delta_\alpha^2$, such that any online control algorithm on a system $\mathcal{L} \sim \mathcal{D}$ incurs expected regret $\Omega(T)$ against the class of all time-invariant linear policies $x \mapsto Kx$ where $K \in \bigcup_{\alpha \in [0, \beta/T]} \mathbb{S}_\alpha^d$.*

## 4 Experimental Evaluation

The previous sections focused on linear systems, but in fact `GPC-Simplex` can be easily modified to control non-linear systems, for similar reasons as in prior work [2]. It suffices for the dynamics to

---

[14]We remark that, with the mixing assumption, Theorem 7 does achieve $\tilde{O}(\sqrt{T})$ regret when $\mathcal{I} := \bigcup_{\alpha \in [0, O(1/T)]} \Delta_\alpha^d$. In particular, there is no hidden dependence on $\mathcal{I}$ in the regret bound.

have the form

$$x_{t+1} := (1 - \gamma_t)f(x_t, u_t) + \gamma_t w_t \tag{9}$$

for known $f$, observed $\gamma_t$, and unknown $w_t$. See Appendix E for discussion of the needed modifications and other implementation details. Relevant code is open-sourced in [16].

As a case study, in this section we apply `GPC-Simplex` (Algorithm 1) to a disease transmission model – specifically, a controlled generalization of the SIR model introduced earlier. In Appendix H we apply `GPC-Simplex` to a controlled version of the *replicator dynamics* from evolutionary game theory.

**A controlled disease transmission model.** The Susceptible-Infectious-Recovered (SIR) model is a basic model for the spread of an epidemic [26]. The SIR model has been extensively studied since last century [36, 41, 4, 24, 6] and attracted renewed interest during the COVID-19 pandemic [10, 29, 9]. As discussed previously, this model posits that a population consists of susceptible (**S**), infected (**I**), and recovered (**R**) individuals. When a susceptible individual comes into contact with an infected individual, the susceptible individual becomes infected at some "transmission rate" $\beta$. Infected patients become uninfected and gain immunity at some "recovery rate" $\theta$. We consider a natural generalization of the standard dynamics Eq. (1) where recovered individuals may also lose immunity at a rate of $\xi$. Formally, in the absence of control, the population evolves over time according to the following system of differential equations:

$$\frac{dS}{dt} = -\beta IS + \xi R, \quad \frac{dI}{dt} = \beta IS - \theta I, \quad \frac{dR}{dt} = \theta I - \xi R, \tag{10}$$

Typically, $\beta > \theta > \xi$. We normalize the total population to be 1, and thus $x = [S, I, R] \in \Delta^3$. Next, we introduce a variable called the *preventative control* $u_t \in \Delta^2$, which has the effect of decreasing the transmission rate $\beta$, and adversarial perturbations $w_t$, which allow for model misspecification. Incorporating these changes to the forward discretization of Eq. (10) gives the following dynamics:

$$\begin{bmatrix} S_{t+1} \\ I_{t+1} \\ R_{t+1} \end{bmatrix} = (1 - \gamma_t) \left( \begin{bmatrix} 1 - \beta I_t & 0 & \xi \\ 0 & 1 - \theta & 0 \\ 0 & \theta & 1 - \xi \end{bmatrix} \begin{bmatrix} S_t \\ I_t \\ R_t \end{bmatrix} + \begin{bmatrix} \beta I_t S_t & 0 \\ 0 & \beta I_t S_t \\ 0 & 0 \end{bmatrix} u_t \right) + \gamma_t w_t. \tag{11}$$

The control $u_t \in \Delta^2$ represents a distribution over transmission prevention protocols: $u_t = [1, 0]$ represents full-scale prevention, whereas $u_t = [0, 1]$ represents that no prevention measure is imposed. Concretely, the effective transmission rate under control $u_t$ is $\beta \cdot u_t(2)$.

**Parameters and cost function.** To model a highly infectious pandemic, we consider Eq. (11) with parameters $\beta = 0.5$, $\theta = 0.03$, and $\xi = 0.005$. Suppose we want to control the number of infected individuals by modulating a (potentially expensive) prevention protocol $u_t$. To model this setting, the cost function includes (1) a quadratic cost for infected individuals $I_t$, and (2) a cost that is bilinear in the magnitude of prevention and the susceptible individuals:

$$c_t(x_t, u_t) = c_3 \cdot x_t(2)^2 + c_2 \cdot x_t(1) \cdot u_t(1), \tag{12}$$

where $x_t = [S_t, I_t, R_t]$. Typically $c_3 \geqslant c_2 > 0$ to model the high cost of infection.

In Fig. 1, we compare `GPC-Simplex` against two baselines – (a) always executing $u_t = [1, 0]$ (i.e. full prevention), and (b) always executing $u_t = [0, 1]$ (i.e. no prevention) – for $T = 200$ steps in the above model with no perturbations. We observe that `GPC-Simplex` suppresses the transmission rate via high prevention at the initial stage of the disease outbreak, then relaxes as the outbreak is effectively controlled. Moreover, `GPC-Simplex` outperforms both baselines in terms of cumulative cost. See Appendix F for additional experiments exhibiting the robustness of `GPC-Simplex` to perturbations (i.e. non-zero $\gamma_t$'s) and different model parameters.

## 4.1 Controlling hospital flows: reproducing a study by [27]

We now turn to the recent work [27], which also studies a controlled SIR model. Similar to above, they considered a control that temporarily reduces the rate of contact within a population. In one scenario (inspired by the COVID-19 pandemic), they considered a cost function that penalizes

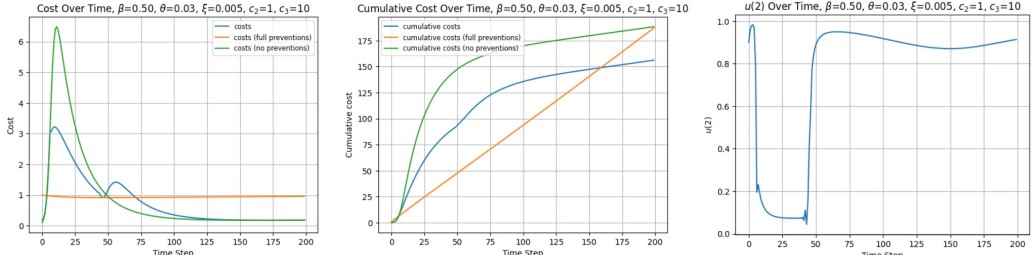

**Figure 1:** Control with cost function (12) for $T = 200$ steps: initial distribution $x_1 = [0.9, 0.1, 0.0]$; parameters $c_3 = 10, c_2 = 1$; no noise. **Left/Middle:** Cost and cumulative cost over time of `GPC-Simplex` versus baselines. **Right:** control $u_t(2)$ (proportional to effective transmission rate) played by `GPC-Simplex` over time.

medical surges, i.e. when the number of infected exceeds a threshold $y_{\max}$ determined by hospital capacities. Formally, they define the cost of a trajectory $(x_t, u_t)_{t=1}^T$ as

$$\frac{W_0(-3x_T(1)e^{-3(x_T(1)+x_T(2))})}{3} + \int_0^T \left[ c_2 \cdot u_t(1)^2 + \frac{c_3(x_t(2) - y_{\max})}{1 + e^{-100(x_t(2) - y_{\max})}} \right] dt, \quad (13)$$

where $W_0$ is the principal branch of Lambert's $W$-function, and $c_2, c_3$ are hyperparameters. The system parameters used by [27] are $\beta = 0.3, \theta = 0.1, \xi = 0$. In the absence of noise and with a known cost function, [27] is able to compute the approximate solutions of the associated Hamilton-Jacobi-Bellman equations for various choices of $c_2, c_3$.

In Fig. 2, we show that `GPC-Simplex` (with a slightly modified instantaneous version of Eq. (13)) in fact *matches* the optimal solution analytically computed by [28]. See Appendix G for further experimental details, including the exact model parameters and cost function.

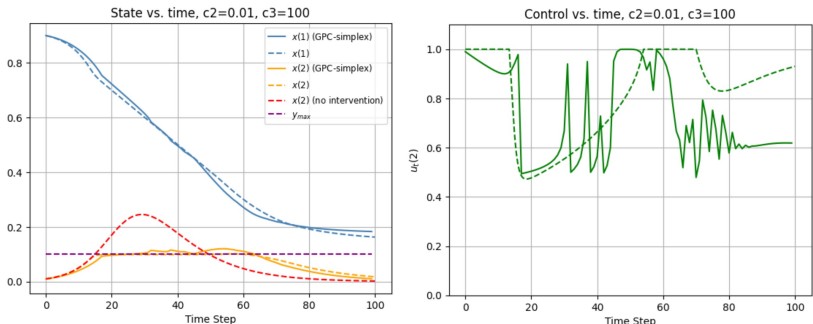

**Figure 2:** Controlling hospital flows for $T = 100$ steps: initial distribution $[0.9, 0.01, 0.09]$; parameters $y_{max} = 0.1, c_2 = 0.01, c_3 = 100$. **Left:** The dashed red line shows the number of infected over time under no control; note that $y_{\max}$ (shown in dashed purple line) is significantly exceeded. The solid yellow and blue lines show the number of infected and susceptible under `GPC-Simplex`, which closely match the optimal solutions computed by [28] (dashed yellow and blue). **Right:** `GPC-Simplex` control (solid) vs. optimal control (dashed).

## Acknowledgements

NG is supported by a Fannie & John Hertz Foundation Fellowship and an NSF Graduate Fellowship. EH, ZL and JS gratefully acknowledge funding from the National Science Foundation, the Office of Naval Research, and Open Philanthropy. DR is supported by a U.S. DoD NDSEG Fellowship.

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

# Contents

## A   Additional preliminaries

For completeness, we recall the definition of a standard LDS [21].

**Definition 9** (LDS). Let $d_x, d_u \in \mathbb{N}$. A *linear dynamical system* (LDS) is described by a tuple $\mathcal{L} = (A, B, x_1, (w_t)_{t \in \mathbb{N}}, (c_t)_{t \in \mathbb{N}})$ where $A \in \mathbb{R}^{d_x \times d_x}$, $B \in \mathbb{R}^{d_x \times d_u}$ are the transition matrices; $x_1 \in \mathbb{R}^{d_x}$ is the initial state; $w_t \in \mathbb{R}^{d_x}$ is the noise value at time $t$; and $c_t : \mathbb{R}^{d_x} \times \mathbb{R}^{d_u} \to \mathbb{R}$ is the cost function at time $t$. For each $t \geqslant 1$, given state $x_t \in \mathbb{R}^{d_x}$ and control $u_t \in \mathbb{R}^{d_u}$ at time $t$, the state at time $t+1$ is given by

$$x_{t+1} := Ax_t + Bu_t + w_t,$$

and the instantaneous cost incurred at time $t$ is given by $c_t(x_t, u_t)$.

## B   Discussion on the observation model

Our main algorithm `GPC-Simplex` for online control of simplex LDSs assumes that for each $t$, the perturbation strength $\gamma_t$ is observed by the controller at the same time as it observes $x_{t+1}$ (the algorithm does not require the entire sequence $(\gamma_t)_t$ to be known in advance). In this appendix we discuss (a) why this is a crucial technical assumption for the algorithm, and (b) why it is a *reasonable* assumption in many natural population models.

First we explain why is it technically important for `GPC-Simplex` that the controller observes $\gamma_t$. Recall that like the algorithm `GPC` from [1], `GPC-Simplex` is a *disturbance-action controller*, meaning that the control at time $t$ is computed based on previous disturbances $w_{t-i}$. In the standard LDS model (Definition 9) studied by [1], it's clear that $w_{t-1}$ can be computed from $x_{t-1}, u_{t-1}, x_t$, using the fact that $A, B$ are known. However, in the simplex LDS model, if $\gamma_{t-1}$ is not directly observed, then in fact $w_{t-1}$ may not be uniquely identifiable given $x_{t-1}, u_{t-1}, x_t$. This is why `GPC-Simplex` requires observing the parameters $\gamma_t$. It is an interesting open problem whether this assumption can be removed.

Second, we argue that in many practical applications, it is reasonable for $\gamma_{t-1}$ to be observed along with the population state $x_t$. The reason is that often the controller can observe not just the proportions of individuals of different categories in a population but also the total population size.

Formally, consider a population which has $N_t$ individuals at time $t$. Thus, if the distribution of the population across $d$ categories is described by $x_t \in \Delta^d$, then for each $i \in [d]$ there are $N_t(x_t)_i$ individuals in category $i$. Suppose that under control $u_t \in \mathcal{I}$, this population evolves to a new distribution $(1 - \|u_t\|_1)Ax_t + Bu_t$, but then the adversary *adds* $n_t$ new individuals to the population, whose distribution over categories is given by $w_t \in \Delta^d$. Then if we write $\bar{x}_t \in \mathbb{R}^d_{\geqslant 0}$ to denote the vector of counts of individuals in each category at time $t$, it holds that

$$\begin{aligned} \bar{x}_{t+1} &= N_t((1 - \|u_t\|_1)Ax_t + Bu_t) + n_t w_t \\ &= N_{t+1}\left((1 - \gamma_t)((1 - \|u_t\|_1)Ax_t + Bu_t) + \gamma_t w_t\right) \end{aligned}$$

where $N_{t+1} = N_t + n_t$ is the total number of individuals at time $t+1$, and we write $\gamma_t := n_t/(N_t + n_t)$. Thus, the distribution of the population across the $d$ categories at time $t+1$ is

$$x_{t+1} = \frac{\bar{x}_{t+1}}{N_{t+1}} = (1 - \gamma_t)((1 - \|u_t\|_1)Ax_t + Bu_t) + \gamma_t w_t$$

which is exactly the update rule from Definition 3. Moreover, if the controller observes the total population counts $N_t, N_{t+1}$ in addition to $x_t, u_t, x_{t+1}$, then it may compute $\gamma_t = (N_{t+1} - N_t)/N_{t+1}$ as well as $w_t$ (using knowledge of $A, B$), which is what we wanted to show.

## C   Proof of Theorem 7

In this section, we prove Theorem 7. We begin with an overview of this section that outlines the structure and the main idea behind the proof of Theorem 7.

**Overview.** GPC-Simplex (Algorithm 1) essentially runs mirror descent on the loss functions $\ell_t(p, M^{[1:H]})$ constructed in Line 9. In particular, the loss at time $t$ measures the counterfactual cost of following the policy $\pi^{p,M^{[1:H]}}$ for the first $t$ timesteps. Thus, the regret of GPC-Simplex against the comparator class $\mathcal{K}_\tau^\triangle(\mathcal{L})$ (Definition 6) can be bounded by the following decomposition:

$$\boxed{\text{Approximation error of comparator class}} + \boxed{\text{Mismatch error of costs}} + \boxed{\texttt{LazyMD regret}}$$

In more detail:

- **Approximation error of comparator class.** Since GPC-Simplex is only optimizing over policies of the form $\pi^{p,M^{[1:H]}}$ for $(p, M^{[1:H]}) \in \mathcal{X}_{d,H,a_0,\overline{\alpha}}$, we must show that every policy in the comparator class can be approximated by some policy $\pi^{p,M^{[1:H]}}$. This is accomplished by Lemma 17.

- **Mismatch error of costs.** The cost incurred by the mirror descent algorithm LazyMD at time $t$ is $\ell_t(p_t, M_t^{[1:H]})$, which is the counterfactual cost at time $t$ had the current policy $\pi^{p_t, M_t^{[1:H]}}$ been carried out from the beginning of the time. However, the cost actually incurred by the controller at time $t$ is $c_t(x_t, u_t)$, which is the cost incurred by following policy $\pi^{p_s, M_s^{[1:H]}}$ at time $t$, for each $s \leqslant t$. Thus, there is a mismatch between the loss that GPC-Simplex is optimizing and the loss that GPC-Simplex needs to optimize. This mismatch can be bounded using the stability of mirror descent along with a mixing argument; see Lemma 21.

- **LazyMD regret.** GPC-Simplex uses LazyMD as its subroutine for mirror descent. The regret of LazyMD can be bounded by standard guarantees; see Corollary 15.

## C.1 Preliminaries on mirror descent

We begin with some preliminaries regarding mirror descent. Let $\mathcal{X} \subset \mathbb{R}^d$ be a convex compact set, and let $R : \mathcal{X} \to \mathbb{R}$ be a convex function. We consider the *Lazy Mirror Descent* algorithm LazyMD (also known as *Following the Regularized Leader*) for online convex optimization on $\mathcal{X}$. Given an offline optimization oracle over $\mathcal{X}$, the function $R$, and a parameter $\eta > 0$, LazyMD chooses each iterate $z_t$ based on the historical loss functions $\ell_s : \mathcal{X} \to \mathbb{R}$ (for $s \in [t-1]$) as described in Algorithm 2.

---

**Algorithm 2** LazyMD: Lazy Mirror Descent [35]

---

**Require:** Offline convex optimization oracle over set $\mathcal{X} \subset \mathbb{R}^d$; convex regularization function $R : \mathcal{X} \to \mathbb{R}$; step size $\eta > 0$; loss functions $\ell_1, \ldots, \ell_T$ where $\ell_t$ is revealed after iteration $t$.
1: **for** $t \geqslant 1$ **do**
2:     Compute and output the solution to the following convex optimization problem:

$$z_t := \arg\min_{z \in \mathcal{X}} \sum_{s=1}^{t-1} \langle z, \nabla \ell_s(z_s) \rangle + \frac{1}{\eta} R(z), \tag{14}$$

3:     Receive loss function $\ell_t : \mathcal{X} \to \mathbb{R}$.

---

The following lemma bounds the regret of LazyMD against the single best $z \in \mathcal{X}$ (in hindsight), for an appropriately chosen step size $\eta$.

**Lemma 10** (Mirror descent)**.** *Suppose that $\mathcal{X} \subset \mathbb{R}^d$ is convex and compact, and let $\|\cdot\|$ be a norm on $\mathbb{R}^d$. Let $R : \mathcal{X} \to \mathbb{R}$ be a 1-strongly convex function with respect to $\|\cdot\|$. Let $L > 0$, and let $\rho := \max_{z \in \mathcal{X}} R(z) - \min_{z \in \mathcal{X}} R(z)$.*

*Fix an arbitrary sequence of loss functions $\ell_t : \mathcal{X} \to \mathbb{R}$ which are each convex and $L$-Lipschitz with respect to $\|\cdot\|$. Then the iterates $z_t$ of LazyMD (Eq. (14)) with an optimization oracle over $\mathcal{X}$, regularizer $R$, step size $\eta = \sqrt{\rho}/(L\sqrt{2T})$, and loss functions $\ell_1, \ldots, \ell_T$ satisfy:*

$$\sum_{t=1}^T \ell_t(z_t) - \min_{z \in \mathcal{X}} \sum_{t=1}^T \ell_t(z) \leqslant L\sqrt{8\rho T} \tag{15}$$

*Moreover, for each $t \in [T-1]$, it holds that*

$$\|z_t - z_{t+1}\| \leqslant \sqrt{\frac{\rho}{2T}}. \tag{16}$$

Lemma 10 is essentially standard but we provide a proof for completeness.

*Proof of Lemma 10.* By [18, Theorem 5.2], it holds that

$$\sum_{t=1}^{T} \ell_t(z_t) - \min_{z \in \mathcal{X}} \ell_t(z) \leqslant 2\eta \sum_{t=1}^{T} \|\nabla \ell_t(z_t)\|_\star^2 + \frac{\rho}{\eta}$$

where $\|\cdot\|_\star : \mathbb{R}^d \to \mathbb{R}$ is the *dual norm* of $\|\cdot\|$, defined by $\|y\|_\star := \max_{\|z\| \leqslant 1} \langle y, z \rangle$. Recall that a convex $L$-Lipschitz loss function $\ell_t$ satisfies $\|\nabla \ell_t(z)\|_\star \leqslant L$ for all $z \in \mathcal{X}$. Thus, the above regret bound simplifies to

$$\sum_{t=1}^{T} \ell_t(z_t) - \min_{z \in \mathcal{X}} \ell_t(z) \leqslant 2\eta T L^2 + \frac{\rho}{\eta}.$$

Substituting in $\eta = \frac{\sqrt{\rho}}{L\sqrt{2T}}$ yields Eq. (15). To establish the movement bound Eq. (16), we argue as follows. Consider any $y_1, y_2 \in \mathbb{R}^d$ and define, for $i \in \{1, 2\}$,

$$w_i := \arg\min_{z \in \mathcal{X}} \langle y_i, z \rangle + R(z).$$

The definition of $w_2$ implies that

$$R(w_1) - \langle y_2, w_2 \rangle + \langle y_2, w_1 \rangle \geqslant R(w_2) \geqslant R(w_1) + \langle \nabla R(w_1), w_2 - w_1 \rangle + \frac{1}{2}\|w_2 - w_1\|^2$$

where the second inequality is by 1-strong convexity of $R$. Simplifying, we get

$$\frac{1}{2}\|w_1 - w_2\|^2 \leqslant \langle y_2, w_1 - w_2 \rangle + \langle \nabla R(w_1), w_2 - w_1 \rangle.$$

Symmetrically, the definition of $w_1$ together with strong convexity implies that

$$\frac{1}{2}\|w_1 - w_2\|^2 \leqslant \langle y_1, w_2 - w_1 \rangle + \langle \nabla R(w_2), w_1 - w_2 \rangle.$$

Adding the two above displays gives

$$\|w_1 - w_2\|^2 \leqslant \langle y_2 - y_1, w_1 - w_2 \rangle + \langle \nabla R(w_1) - \nabla R(w_2), w_2 - w_1 \rangle$$
$$\leqslant \|y_2 - y_1\|_\star \cdot \|w_1 - w_2\|,$$

where the second inequality uses convexity of $R$ (which gives $\langle \nabla R(w_1) - \nabla R(w_2), w_1 - w_2 \rangle \geqslant 0$). It follows that $\|w_1 - w_2\| \leqslant \|y_1 - y_2\|_\star$. Setting $y_1 := \eta \sum_{s=1}^{t-1} \nabla \ell_s(z_s)$ and $y_2 := \eta \sum_{s=1}^{t} \nabla \ell_s(z_s)$, and recalling the definitions of $z_t, z_{t+1}$ from Eq. (14), we get

$$\|z_t - z_{t+1}\| \leqslant \|\eta \nabla \ell_t(z_t)\|_\star \leqslant \eta L \leqslant \sqrt{2\rho/T},$$

as desired. $\qquad \square$

We next apply Lemma 10 to the domain used in `GPC-Simplex`. Recall that, given $d, H \in \mathbb{N}$ and real numbers $0 \leqslant a \leqslant b \leqslant 1$, we have defined $\mathcal{X}_{d,H,a,b} := \bigcup_{a' \in [a,b]} \Delta_{a'}^d \times (\mathbb{S}_{a'}^d)^H$.

**Definition 11** (Entropy of a sub-distribution). Let $d \in \mathbb{N}$. We define the function $\mathrm{Ent} : \Delta_{\leqslant 1}^d \to \mathbb{R}_{\geqslant 0}$ by

$$\mathrm{Ent}(v) := v^{\mathrm{c}} \ln \frac{1}{v^{\mathrm{c}}} + \sum_{j=1}^{d} v_j \ln \frac{1}{v_j}$$

where for any vector $v \in \mathbb{R}^d$ we write $v^{\mathrm{c}} := 1 - \sum_{j=1}^{d} v_j \in \mathbb{R}$.

**Lemma 12.** *Let $d \in \mathbb{N}$ and $u, v \in \Delta_{\leqslant 1}^d$. Then*

$$\langle \nabla_u \operatorname{Ent}(u) - \nabla_v \operatorname{Ent}(v), u - v \rangle \leqslant - \|u - v\|_1^2.$$

*That is, $v \mapsto - \operatorname{Ent}(v)$ is 1-strongly convex on $\Delta_{\leqslant 1}^d$ with respect to $\|\cdot\|_1$.*

*Proof.* Let $p$ be the probability mass function on $[d+1]$ with $p_i = u_i$ for all $i \in [d]$, and let $q$ be the probability mass function on $[d+1]$ with $p_i = v_i$ for all $i \in [d]$. Then it can be checked that

$$\langle \nabla_u \operatorname{Ent}(u) - \nabla_v \operatorname{Ent}(v), v - u \rangle = \operatorname{KL}(p\|q) + \operatorname{KL}(q\|p)$$
$$\geqslant \operatorname{TV}(p, q)^2$$
$$\geqslant \|u - v\|_1^2$$

where the first inequality is by Pinsker's inequality. $\qquad \square$

**Definition 13** (Regularizer for mirror descent in `GPC-Simplex`). Let $d, H \in \mathbb{N}$ and $0 \leqslant a \leqslant b \leqslant 1$. We define $R_{d,H} : \mathcal{X}_{d,H,a,b} \to \mathbb{R}_{\leqslant 0}$ as follows (omitting the domain's dependence on $a, b$ for notational simplicity):

$$R_{d,H}(p, M^{[1:H]}) := - \operatorname{Ent}(p) - \sum_{h=1}^{H} \sum_{j=1}^{d} \operatorname{Ent}(M_{\cdot,j}^{[h]})., \tag{17}$$

**Definition 14** (Norm for analysis of mirror descent in `GPC-Simplex`). Let $d, H \in \mathbb{N}$, and identify $\mathbb{R}^{d+Hd^2}$ with $\mathbb{R}^d \times (\mathbb{R}^{d \times d})^H$. We define a norm $\|\cdot\|_{d,H}$ on $\mathbb{R}^{d+Hd^2}$ as follows: for $p \in \mathbb{R}^d, M^{[1:H]} \in (\mathbb{R}^{d \times d})^H$,

$$\left\| (p, M^{[1:H]}) \right\|_{d,H}^2 := \|p\|_1^2 + \sum_{h=1}^{H} \sum_{j=1}^{d} \left\| M_{\cdot,j}^{[h]} \right\|_1^2.$$

**Corollary 15.** *Let $d, H \in \mathbb{N}$ and $0 \leqslant a \leqslant b \leqslant 1$. Consider an arbitrary sequence of cost functions $\ell_t : \mathcal{X}_{d,H,a,b} \to \mathbb{R}$ which are convex and $L$-Lipschitz with respect to $\|\cdot\|_{d,H}$. Then the iterates $z_t$ of `LazyMD` with $\eta = \sqrt{2dH \ln(d)}/(L\sqrt{T})$, regularizer $R$, and loss functions $\ell_1, \ldots, \ell_T$ satisfy the following regret guarantee:*

$$\sum_{t=1}^{T} \ell_t((p_t, M_t^{[1:H]})) - \min_{(p, M^{[1:H]}) \in \mathcal{X}_{d,H,a,b}} \sum_{t=1}^{T} \ell_t((p, M^{[1:H]})) \leqslant L\sqrt{32dH \ln(d) \cdot T} \tag{18}$$

*Moreover, for $\beta := \frac{\sqrt{2dH \ln(d)}}{\sqrt{T}}$, for all $t \in [T-1]$, we have*

$$\|p_t - p_{t+1}\|_1 \leqslant \beta, \qquad \max_{h \in [H]} \left\| M_t^{[h]} - M_{t+1}^{[h]} \right\|_{1 \to 1} \leqslant \beta. \tag{19}$$

*Proof.* Note that the set of $(p, M^{[1:H]})$ where $p \in \mathbb{R}^d$ and $M^{[1:H]} \in (\mathbb{R}^{d \times d})^H$ can be identified with $\mathbb{R}^{d+Hd^2}$. We apply Lemma 10 with $\mathcal{X} := \mathcal{X}_{d,H,a,b}$, $R = R_{d,H}$, and the norm $\|\cdot\|_{d,H}$. It is straightforward to check that $\mathcal{X}$ is convex and compact in $\mathbb{R}^{d+Hd^2}$. By Lemma 12, we have that $R_{d,H}$ is 1-strongly convex with respect to the norm $\|\cdot\|_{d,H}$. Moreover, note that

$$\max_{(p, M^{[1:H]}) \in \mathcal{X}_{d,H,a,b}} R_{d,H}((p, M^{[1:H]})) - \min_{(p, M^{[1:H]}) \in \mathcal{X}_{d,H,a,b}} R_{d,H}((p, M^{[1:H]}))$$
$$\leqslant (1 + dH) \ln(d+1)$$
$$\leqslant 4dH \ln(d)$$

since $0 \leqslant \operatorname{Ent}(v) \leqslant \ln(d+1)$ for all $v \in \Delta_{\leqslant 1}^d$. Thus, Lemma 10 implies the claimed bounds Eqs. (18) and (19), where to prove Eq. (19) we are using the fact that $\|C\|_{1 \to 1} = \max_{j \in [d]} \|C_{\cdot,j}\|_1 \leqslant \sqrt{\sum_{j \in [d]} \|C_{\cdot,j}\|_1^2}$ for all $C \in \mathbb{R}^{d \times d}$. $\qquad \square$

## C.2 Approximation of linear policies

Henceforth fix a simplex LDS $\mathcal{L} = (A, B, \mathcal{I}, x_1, (\gamma_t)_t, (w_t)_t, (c_t)_t)$ on $\Delta^d$, where $\mathcal{I} = \bigcup_{\alpha \in [\underline{\alpha}, \overline{\alpha}]} \Delta^d_\alpha$ for some constants $0 \leqslant \underline{\alpha} \leqslant \overline{\alpha} \leqslant 1$.

Recall that any choice of parameters $(p, M^{[1:H]}) \in \mathcal{X}_{d,H,a,b}$ (for some hyperparameters $H \in \mathbb{N}$ and $\underline{\alpha} \leqslant a \leqslant b \leqslant \overline{\alpha}$) induces, via Eq. (8), a set of policies $(\pi_s^{p,M^{[1:H]}})_{s \in [T]}$. The policy $\pi_s^{p,M^{[1:H]}}$ takes as input the disturbances $w_{s-1}, \ldots, w_{s-H}$ observed at the $H$ time steps before step $s$, and outputs a control for step $s$. Recall that, in Algorithm 1, we used $x_t(p, M^{[1:H]}), u_t(p, M^{[1:H]})$ to denote the state and control at time step $t$ one would observe by playing the control $\pi_s^{p,M^{[1:H]}}(w_{s-1:s-H})$ at step $s$, for each $1 \leqslant s \leqslant t$.

Formally, we have the following expressions for $x_t(p, M^{[1:H]}), u_t(p, M^{[1:H]})$:

**Fact 16.** *For any* $(p, M^{[1:H]}) \in \mathcal{X}_{d,H,a,b}$ *and* $t \in [T]$*, it holds that*

$$x_t(p, M^{[1:H]}) = \sum_{i=1}^{t} \alpha_{t,i}^{p,M^{[1:H]}} \cdot A^{i-1} \cdot \left( \lambda_{t-i,0} \bar{\lambda}_{t,i} \cdot B \cdot p + B \sum_{j=1}^{H} \lambda_{t,i+j} \cdot M^{[j]} \cdot w_{t-i-j} + \lambda_{t,i} \cdot w_{t-i} \right)$$
(20)

$$u_t(p, M^{[1:H]}) = \pi_t^{p,M^{[1:H]}}(w_{t-1:t-H}) = \lambda_{t,0} \cdot p + \sum_{j=1}^{H} \lambda_{t,j} \cdot M^{[j]} \cdot w_{t-j},$$
(21)

*where we have written, for* $t \in [T], i \in \mathbb{N}$*,* $\alpha_{t,i}^{p,M^{[1:H]}} := \prod_{j=1}^{i-1} \left( 1 - \left\| \pi_{t-j}^{p,M^{[1:H]}} \right\|_1 \right) = (1 - \|p\|_1)^{i-1}$.

*Proof.* This follows unrolling Eq. (6) with the controls $u_s := \pi_s^{p,M^{[1:H]}}(w_{s-1:s-H})$, and recalling the definitions in Eq. (7) and the conventions $w_0 := x_1, \gamma_0 = 1$, and $w_t = 0$ for $t < 0$. $\square$

In a sense, Algorithm 1 performs online convex optimization over the set of such policies. Even if we can manage to show that doing so yields a good regret guarantee with respect to the class of policies $\{\pi_t^{p,M^{[1:H]}} : (p, M^{[1:H]}) \in \mathcal{X}_{d,H,a,b}\}$ for some choices of $H, a, b$, why should this imply a good regret guarantee with respect to the class $\mathcal{K}_\tau^\triangle(\mathcal{L})$ of linear policies (see Definition 6)? Lemma 17 bridges this gap, showing that any policy in $\mathcal{K}_\tau^\triangle(\mathcal{L})$ can be approximated by a policy of the form $(\pi_s^{p,M^{[1:H]}})_{s \in [T]}$.

**Lemma 17** (Approximation). *Suppose that the cost functions* $c_1, \ldots, c_T$ *of* $\mathcal{L}$ *satisfy Assumption 1 with Lipschitz parameter $L$. Fix* $\tau > 0$, $\varepsilon \in (0,1)$, *and any* $K^\star \in \mathbb{S}^d_{\leqslant 1}$ *such that* $t^{\text{mix}}(\mathbb{A}_{K^\star}) \leqslant \tau$. *Write* $\alpha^\star := \|K^\star\|_{1 \to 1}$. *If* $H \geqslant \tau \lceil \log_2(2LT^2/\varepsilon) \rceil$*, then there is some* $(p, M^{[1:H]}) \in \Delta^d_{\alpha^\star} \times (\mathbb{S}^d_{\alpha^\star})^H$ *such that*

$$\sum_{t=1}^{T} c_t(x_t(p, M^{[1:H]}), u_t(p, M^{[1:H]})) - \sum_{t=1}^{T} c_t(x_t(K^\star), u_t(K^\star)) \leqslant \varepsilon,$$
(22)

*where* $x_t(K^\star), u_t(K^\star)$ *denote the state and control that one would observe at time step $t$ if one were to play according to the policy* $x \mapsto K^\star x$ *at all time steps* $1 \leqslant s \leqslant t$.

*Proof.* For each $t$, if the controls $u_t$ are chosen to satisfy $u_t := K^\star \cdot x_t$, then we have $\alpha_t := \|K^\star\|_{1 \to 1}$. Moreover, for $1 \leqslant t \leqslant T$, we can write

$$x_t(K^\star) = \sum_{i=1}^{t} (\mathbb{A}_{K^\star})^{i-1} \cdot \left( \prod_{j=1}^{i-1} (1 - \gamma_{t-j}) \right) \cdot \gamma_{t-i} w_{t-i} = \sum_{i=1}^{t} \mathbb{A}_{K^\star}^{i-1} \cdot \lambda_{t,i} \cdot w_{t-i},$$
(23)

$$u_t(K^\star) = K^\star \cdot x_t(K^\star) = \sum_{i=1}^{t} K^\star \mathbb{A}_{K^\star}^{i-1} \cdot \lambda_{t,i} \cdot w_{t-i}$$
(24)

where $\mathbb{A}_{K^\star}$ was defined in Eq. (5). By the assumption that $t^{\text{mix}}(\mathbb{A}_{K^\star}) \leqslant \tau$, there is some unique $p' \in \Delta^d$ such that that $\mathbb{A}_{K^\star} \cdot p' = p'$ (see Definition 5). Moreover, by our bound on $H$ and

Lemma 18, for any $i > H$ and $q \in \Delta^d$ we have $\left\| \mathbb{A}_{K^\star}^{i-1} q - p' \right\|_1 \leq (1/2)^{H/\tau} \leq \varepsilon/(2LT^2)$. Using that $\lambda_{t,0} = \sum_{i=H+1}^{t} \lambda_{t,i}$ by the definition in Eq. (7),

$$\left\| \sum_{i=H+1}^{t} K^\star \mathbb{A}_{K^\star}^{i-1} \lambda_{t,i} w_{t-i} - \lambda_{t,0} \cdot K^\star p' \right\|_1 = \left\| K^\star \sum_{i=H+1}^{t} \lambda_{t,i} (\mathbb{A}_{K^\star}^{i-1} w_{t-i} - p') \right\|_1$$

$$\leq \sum_{i=H+1}^{t} \lambda_{t,i} \cdot \left\| \mathbb{A}_{K^\star}^{i-1} w_{t-i} - p' \right\|_1 \leq \varepsilon/(2LT^2). \quad (25)$$

For $1 \leq i \leq H$, let us define $M^{[i]} := K^\star \mathbb{A}_{K^\star}^{i-1} \in \mathbb{S}_{\alpha^\star}^d$ and $p := K^\star \cdot p' \in \Delta_{\alpha^\star}^d$. Using Eqs. (21) and (24), we have that

$$\left\| u_t(K^\star) - u_t(p, M^{[1:H]}) \right\|_1 = \left\| \sum_{i=1}^{t} K^\star \mathbb{A}_{K^\star}^{i-1} \lambda_{t,i} w_{t-i} - \lambda_{t,0} p - \sum_{j=1}^{H} \lambda_{t,j} M^{[j]} w_{t-j} \right\|_1$$

$$= \left\| \sum_{i=H+1}^{t} K^\star \mathbb{A}_{K^\star}^{i-1} \lambda_{t,i} w_{t-i} - \lambda_{t,0} \cdot K^\star p' \right\|_1$$

$$\leq \varepsilon/(2LT^2), \quad (26)$$

where the final inequality uses Eq. (25).

Next, we may bound the difference in state vectors using Eq. (26), as follows: for any sequence of $(u_i)_{i=1}^{t}$ with $\|u_i\|_1 = \alpha^\star$ for all $i$, we can expand Eq. (6) to get

$$x_t = \sum_{i=1}^{t} (1 - \alpha^\star)^{i-1} A^{i-1} (\bar\lambda_{t,i} B u_{t-i} + \lambda_{t,i} w_{t-i}).$$

Thus, for any $t \in [T]$, we have

$$\left\| x_t(K^\star) - x_t(p, M^{[1:H]}) \right\|_1 \leq \sum_{i=1}^{t} (1 - \alpha^\star)^{i-1} \bar\lambda_{t,i} \cdot \left\| A^{i-1} B \cdot \left( u_{t-i}(K^\star) - u_{t-i}(p, M^{[1:H]}) \right) \right\|_1$$

$$\leq \frac{\varepsilon}{2LT^2} \cdot \sum_{i=1}^{t} \bar\lambda_{t,i}$$

$$\leq \frac{\varepsilon}{2LT}. \quad (27)$$

By Eqs. (26) and (27) and Assumption 1, it follows that, for each $t \in [T]$,

$$\left| c_t(x_t(p, M^{[1:H]}), u_t(p, M^{[1:H]})) - c_t(x_t(K^\star), u_t(K^\star)) \right| \leq \varepsilon/T,$$

which yields the claimed bound Eq. (22). $\qquad\square$

The following facts about distance to stationarity are well-known (see e.g. [31, Section 4.4]):

**Lemma 18.** *Let $X \in \mathbb{S}^d$ have a unique stationary distribution $\pi$. Then the following inequalities hold for any $c, t \in \mathbb{N}$:*

1. $D_X(t) \leq \bar{D}_X(t) \leq 2D_X(t)$.
2. $\bar{D}_X(ct) \leq \bar{D}_X(t)^c$.

### C.3 Bounding the memory mismatch error

In this section, we prove Lemma 21, which allows us to show that an algorithm with bounded aggregate loss with respect to the loss functions $\ell_t$ defined on Line 9 of Algorithm 1 in fact has bounded aggregate cost with respect to the cost functions $c_t$ chosen by the adversary.

First, we introduce two useful lemmas on the mixing time of matrices (Definition 5).

**Lemma 19.** *Let $X \in \mathbb{S}^d$ have a unique stationary distribution. Let $Y \in \mathbb{S}^d$ satisfy $\|X - Y\|_{1 \to 1} \leq \delta$. Then for any $t \in \mathbb{N}$,*

$$D_Y(t) \leq 2t\delta + 2D_X(t).$$

*Proof.* For any $v \in \Delta^d$, we have $\|Xv - Yv\|_1 \leqslant \delta$. A hybrid argument then yields that for any $t \geqslant 1$, $\|X^t v - Y^t v\|_1 \leqslant t\delta$. Then

$$\bar{D}_Y(t) \leqslant \sup_{p,q \in \Delta^d} \|Y^t(p-q)\|_1 \leqslant 2t\delta + \sup_{p,q \in \Delta^d} \|X^t(p-q)\|_1 \leqslant 2t\delta + \bar{D}_X(t) \leqslant 2t\delta + 2D_X(t),$$

where the first and last inequalities apply the first item of Lemma 18. $\qquad\square$

**Lemma 20.** *Suppose that $A, B \in \mathbb{S}^d$, $K^\star \in \mathbb{S}^d_{\leqslant 1}$ satisfy $t^{\mathrm{mix}}(A) > 4 \cdot t^{\mathrm{mix}}(\mathbb{A}_{K^\star})$. Then $\|K^\star\|_{1 \to 1} > 1/(96 \cdot t^{\mathrm{mix}}(\mathbb{A}_{K^\star}))$.*

*Proof.* Let us write $\tau := t^{\mathrm{mix}}(\mathbb{A}_{K^\star})$ and $\alpha^\star := \|K^\star\|_{1 \to 1}$, so that $\mathbb{A}_{K^\star} = (1 - \alpha^\star) \cdot A + BK^\star$. Suppose for the purpose of contradiction that $\alpha^\star \leqslant 1/(96\tau)$. We have that $\|A - \mathbb{A}_{K^\star}\|_{1 \to 1} \leqslant 2\alpha^\star$. By Lemma 18 and Definition 5, we have $\bar{D}_{\mathbb{A}_{K^\star}}(\tau) \leqslant 2D_{\mathbb{A}_{K^\star}}(\tau) \leqslant 1/2$, so $D_{\mathbb{A}_{K^\star}}(4\tau) \leqslant \bar{D}_{\mathbb{A}_{K^\star}}(4\tau) \leqslant 1/16$. Using Lemma 19 and the assumption on $\alpha^\star$,

$$D_A(4\tau) \leqslant 12\tau\alpha^\star + 2D_{\mathbb{A}_{K^\star}}(4\tau) \leqslant 12\tau\alpha^\star + 1/8 \leqslant 1/4,$$

meaning that $t^{\mathrm{mix}}(A) \leqslant 4\tau$. $\qquad\square$

The last step is to bound the memory mismatch error.

**Lemma 21** (Memory mismatch error). *Suppose that $(c_t)_t$ satisfy Assumption 1 with Lipschitz parameter $L$. Let $\tau, \beta > 0$, and suppose that $\mathcal{K}^\triangle_\tau(\mathcal{L})$ is nonempty. Consider the execution of* GPC-Simplex *(Algorithm 1) on $\mathcal{L}$ with input $\tau$. If the iterates $(p_t, M_t^{[1:H]})_{t \in [T]}$ satisfy*

$$\|p_t - p_{t+1}\|_1 \leqslant \beta, \qquad \max_{i \in [H]} \left\|M_t^{[i]} - M_{t+1}^{[i]}\right\|_{1 \to 1} \leqslant \beta, \tag{28}$$

*then for each $t \in [T]$, the loss function $\ell_t$ computed at time step $t$ satisfies*

$$|\ell_t(p_t, M_t^{[1:H]}) - c_t(x_t, u_t)| \leqslant O\left(L\tau^3 \beta \log^3(1/\beta)\right).$$

*Proof.* Recall that $u_t \in \Delta^d$ denotes the control chosen in step $t$ of Algorithm 1. We write $\alpha_t := \|u_t\|_1$ and, for $i \in [t]$, $\alpha_{t,i} := \prod_{j=1}^{i-1}(1 - \alpha_{t-j})$. Note that $\alpha_t = \|p_t\|_1 = \left\|M_t^{[h]}\right\|_{1 \to 1}$ for each $h \in [H]$, by definition of $\mathcal{X}_{d,H,a_0,\bar{\alpha}}$.

Let us fix $t \in [T]$, and write $p := p_t, M^{[1:H]} := M_t^{[1:H]}$. By Eq. (6), the state $x_t$ at step $t$ of Algorithm 1 can be written as follows:

$$x_t = \sum_{i=1}^t \alpha_{t,i} \cdot A^{i-1} \cdot \left(\lambda_{t-i,0}\bar{\lambda}_{t,i}Bp_{t-i} + B\sum_{j=1}^H M_{t-i}^{[j]}\lambda_{t,i+j}w_{t-i-j} + \lambda_{t,i}w_{t-i}\right). \tag{29}$$

By assumption that $\mathcal{K}^\triangle_\tau(\mathcal{L})$ is nonempty, there is some $K^\star \in \mathbb{S}^d_{[\underline{\alpha},\overline{\alpha}]}$ satisfying $t^{\mathrm{mix}}(\mathbb{A}_{K^\star}) \leqslant \tau$. Let us write $\alpha^\star := \|K^\star\|_{1 \to 1}$, so that $\mathbb{A}_{K^\star} = (1 - \alpha^\star)A + BK^\star$. Moreover, recall we have written in Algorithm 1 that $\tau_A := t^{\mathrm{mix}}(A)$.

For $1 \leqslant i \leqslant t$, define

$$v_i := \lambda_{t-i,0}\bar{\lambda}_{t,i}Bp_{t-i} + B\sum_{j=1}^H M_{t-i}^{[j]}\lambda_{t,i+j}w_{t-i-j} + \lambda_{t,i}w_{t-i},$$

$$v_i' := \lambda_{t-i,0}\bar{\lambda}_{t,i}Bp + B\sum_{j=1}^H M^{[j]}\lambda_{t,i+j}w_{t-i-j} + \lambda_{t,i}w_{t-i}.$$

Note that

$$\max\{\|v_i\|_1, \|v_i'\|_1, \|v_i - v_i'\|_1\} \leqslant \lambda_{t-i,0}\bar{\lambda}_{t,i} + \sum_{j=1}^H \lambda_{t,i+j} + \lambda_{t,i} \leqslant 1. \tag{30}$$

Next, using Eq. (29) and Eq. (20), we have

$$x_t - x_t(p, M^{[1:H]}) = \sum_{i=1}^{t} \left( \alpha_{t,i} \cdot A^{i-1} \cdot v_i - \alpha_{t,i}^{p,M^{[1:H]}} \cdot A^{i-1} \cdot v_i' \right). \tag{31}$$

The condition Eq. (28) together with the triangle inequality gives that $\|Bp_{-i} - Bp\|_1 \leqslant i\beta$ and $\left\| BM_{t-i}^{[j]} w_{t-i-j} - BM^{[j]} w_{t-i-j} \right\|_1 \leqslant i\beta$ for all $i, j \geqslant 1$, as well as $|\alpha_{t-i} - \alpha_t| \leqslant i\beta$ for all $i \geqslant 1$.
It follows that $\|v_i - v_i'\|_1 \leqslant i\beta$ and $|\alpha_{t,i} - \alpha_{t,i}^{p,M^{[1:H]}}| \leqslant i^2\beta$ for all $i \geqslant 1$ and that for any $\ell \geqslant 1$,

$$\left| \sum_{i=1}^{\ell} \alpha_{t,i} \cdot \|v_i\|_1 - \sum_{i=1}^{\ell} \alpha_{t,i}^{p,M^{[1:H]}} \cdot \|v_i'\|_1 \right| \leqslant \left| \sum_{i=1}^{\ell} |\alpha_{t,i} - \alpha_{t,i}^{p,M^{[1:H]}}| \cdot \|v_i\|_1 \right| + \sum_{i=1}^{\ell} \alpha_{t,i}^{p,M^{[1:H]}} \cdot \|v_i - v_i'\|_1$$
$$\leqslant \ell^3 \beta. \tag{32}$$

Using Eq. (32) and the fact that $\sum_{i=1}^{t} \alpha_{t,i} \|v_i\|_1 = \sum_{i=1}^{t} \alpha_{t,i}^{p,M^{[1:H]}} \|v_i'\|_1 = 1$, we see

$$\left| \sum_{i=\ell+1}^{t} \alpha_{t,i} \cdot \|v_i\|_1 - \sum_{i=\ell+1}^{t} \alpha_{t,i}^{p,M^{[1:H]}} \cdot \|v_i'\|_1 \right| \leqslant \ell^3 \beta. \tag{33}$$

We consider the following two cases:

**Case 1: $\tau_A \leqslant 4\tau$.** Write $t_0 = \lfloor \tau_A \log_2(1/\beta) \rfloor$. Let the stationary distribution of $A$ be denoted $p^\star \in \Delta^d$. By Lemma 18, we have that for all $i \geqslant 1$, $\|A^i \cdot p - p^\star\|_1 \leqslant D_A(i) \leqslant 1/2^{\lfloor i/\tau_A \rfloor}$. Now, using Eq. (31), we may compute

$$\left\| x_t - x_t(p, M^{[1:H]}) \right\|_1$$
$$\leqslant \left\| \sum_{i=1}^{t_0} \left( \alpha_{t,i} \cdot A^{i-1} \cdot v_i - \alpha_{t,i}^{p,M^{[1:H]}} \cdot A^{i-1} \cdot v_i' \right) \right\|_1$$
$$\quad + \left\| \sum_{i=t_0+1}^{t} \alpha_{t,i} \cdot \left( A^{i-1} \cdot v_i - \|v_i\|_1 \cdot p^\star \right) - \alpha_{t,i}^{p,M^{[1:H]}} \cdot \left( A^{i-1} \cdot v_i' - \|v_i'\|_1 \cdot p^\star \right) \right\|_1$$
$$\quad + \left\| \sum_{i=t_0+1}^{t} \alpha_{t,i} \cdot \|v_i\|_1 \cdot p^\star - \alpha_{t,i}^{p,M^{[1:H]}} \cdot \|v_i'\|_1 \cdot p^\star \right\|_1$$
$$\leqslant \sum_{i=1}^{t_0} \left( \alpha_{t,i} \cdot \|A^{i-1} \cdot (v_i - v_i')\|_1 + |\alpha_{t,i} - \alpha_{t,i}^{p,M^{[1:H]}}| \cdot \|v_i'\|_1 \right)$$
$$\quad + \sum_{i=t_0+1}^{t} \left( \alpha_{t,i} \cdot \|A^{i-1} \cdot v_i - \|v_i\|_1 \cdot p^\star\|_1 + \alpha_{t,i}^{p,M^{[1:H]}} \cdot \|A^{i-1} \cdot v_i' - \|v_i'\|_1 \cdot p^\star\|_1 \right) + t_0^3 \beta$$
$$\leqslant t_0^3 \beta + \sum_{i=1}^{t_0} i^2 \beta + \sum_{i=1}^{t_0} \alpha_{t,i} \cdot i\beta + \sum_{i=1+t_0}^{t} 2 \cdot 1/2^{\lfloor i/\tau_A \rfloor}$$
$$\leqslant C t_0^3 \beta \tag{34}$$
$$\leqslant C' \tau^3 \log^3(1/\beta) \cdot \beta,$$

for some universal constants $C, C'$. Above, the first inequality uses the triangle inequality, the second inequality uses Eq. (33), and the third inequality uses that $\|v_i - v_i'\|_1 \leqslant i\beta$, $|\alpha_{t,i} - \alpha_{t,i}^{p,M^{[1:H]}}| \leqslant i^2\beta$, $\|v_i'\|_1 \leqslant 1$. The fourth inequality uses the bound $\sum_{i=1+t_0}^{t} 2^{-\lfloor i/\tau_A \rfloor} \leqslant O(\tau_A \beta) \leqslant O(t_0 \beta)$.

**Case 2: $\tau_A > 4\tau$.** In this case, we claim that $a_0 \geqslant 1/(96\tau)$. By choice of $a_0$ in Line 1 of Algorithm 1 and the fact that $\tau_A > 4\tau$, it suffices to show that $\overline{\alpha} \geqslant 1/(96\tau)$: to see this, note that $\tau_A = t^{\text{mix}}(A) > 4\tau \geqslant 4 \cdot t^{\text{mix}}(\mathbb{A}_{K^\star})$, so Lemma 20 gives that $\|K^\star\|_{1 \to 1} > 1/(96 \cdot t^{\text{mix}}(\mathbb{A}_{K^\star})) \geqslant 1/(96\tau)$. But $\|K^\star\|_{1 \to 1} \leqslant \overline{\alpha}$, and thus $\overline{\alpha} > 1/(96\tau)$. This proves that $a_0 \geqslant 1/(96\tau)$. Hence $\alpha_i \geqslant a_0 \geqslant 1/(96\tau)$, by definition of $\mathcal{X}_{d,H,a_0,\overline{\alpha}}$, for all $i \in [T]$.

Write $t_0 := \lfloor 200\tau \cdot \log(1/\beta) \rfloor$. Then for any $i > t_0$,

$$\max\{\alpha_{t,i}, \alpha_{t,i}^{p,M^{[1:H]}}\} \leqslant (1-a_0)^{i-1} \leqslant (1 - 1/(96\tau))^{\lfloor 200\tau \cdot \log(1/\beta) \rfloor} \leqslant O(\beta).$$

Again using Eq. (31),

$$
\begin{aligned}
\left\| x_t - x_t(p, M^{[1:H]}) \right\|_1 &\leqslant \sum_{i=1}^{t_0} \left( |\alpha_{t,i} - \alpha_{t,i}^{p,M^{[1:H]}}| + \alpha_{t,i} \cdot \|v_i - v_i'\|_1 \right) + \sum_{i=t_0+1}^{t} (\alpha_{t,i} + \alpha_{t,i}^{p,M^{[1:H]}}) \\
&\leqslant \sum_{i=1}^{t_0} \left( i^2 \beta + i\beta \right) + \sum_{i=t_0+1}^{t} O(\beta) \cdot (1-a_0)^{i-t_0-1} \qquad (35) \\
&\leqslant C t_0^3 \beta + C\beta/a_0 \\
&\leqslant C' \tau^3 \log^3(1/\beta) \cdot \beta,
\end{aligned}
$$

for some constants $C, C'$. Above, the first inequality uses Eq. (30); the second inequality uses the previously derived bounds $|\alpha_{t,i} - \alpha_{t,i}^{p,M^{[1:H]}}| \leqslant i^2 \beta$ and $\|v_i - v_i'\|_1 \leqslant i\beta$; and the final inequality uses that $a_0 \geqslant 1/(96\tau)$.

In both cases, we have $\left\| x_t - x_t(p, M^{[1:H]}) \right\|_1 \leqslant C' \tau^3 \beta \log^3(1/\beta)$ for some universal constant $C'$. By definition, the control $u_t$ chosen by Algorithm 1 at time step $t$ is exactly $u_t = u_t(p, M^{[1:H]})$. Thus, using $L$-Lipschitzness of $c_t$, we have

$$
\begin{aligned}
\left| \ell_t(p, M^{[1:H]}) - c_t(x_t, u_t) \right| &= \left| c_t(x_t(p, M^{[1:H]}), u_t(p, M^{[1:H]})) - c_t(x_t, u_t) \right| \\
&\leqslant L \cdot \left\| x_t - x_t(p, M^{[1:H]}) \right\|_1 \\
&\leqslant C' L \tau^3 \beta \log^3(1/\beta).
\end{aligned}
$$

as desired. $\qquad \square$

## C.4 Proof of Theorem 7

Before proving Theorem 7, we establish that the loss functions $\ell_t$ used in GPC-Simplex are Lipschitz.

**Lemma 22.** *Let $X \in \mathbb{S}^d$ with $\tau := t^{\mathsf{mix}}(X) < \infty$. Then for any $i \in \mathbb{N}$ and $v \in \mathbb{R}^d$ with $\langle \mathbb{1}, v \rangle = 0$, it holds that $\left\| X^i v \right\|_1 \leqslant 2^{-\lfloor i/\tau \rfloor} \|v\|_1$.*

*Proof.* Fix $v \in \mathbb{R}^d$ with $\langle \mathbb{1}, v \rangle = 0$. We can write $v = v^+ - v^-$, where $v^+, v^- \in \mathbb{R}_{\geqslant 0}^d$ are the non-negative and negative components of $v$ respectively. We have $\|v^+\|_1 = \|v^-\|_1 = \frac{1}{2}\|v\|_1$ since $\langle \mathbb{1}, v \rangle = 0$. Let $u_1 := 2v^+/\|v\|_1$ and $u_2 := 2v^-/\|v\|_1$, so that $u_1, u_2 \in \Delta^d$. By Lemma 18 and the definition of $t^{\mathsf{mix}}(X)$, we have

$$\|X^i(u_1 - u_2)\|_1 \leqslant \bar{D}_X(i) \leqslant \bar{D}_X(\tau)^{\lfloor i/\tau \rfloor} \leqslant (2D_X(\tau))^{\lfloor i/\tau \rfloor} \leqslant 2^{-\lfloor i/\tau \rfloor}.$$

Thus, $\|X^\tau v\|_1 \leqslant 2^{-\lfloor i/\tau \rfloor} \|v\|_1$. $\qquad \square$

**Lemma 23** (Lipschitzness of $\ell_t$). *Let $\tau > 0$, and suppose that $\mathcal{K}_\tau^{\triangle}(\mathcal{L})$ is nonempty. For each $t \in [T]$, the loss function $\ell_t(p, M^{[1:H]}) = c_t(x_t(p, M^{[1:H]}), u_t(p, M^{[1:H]}))$ (as defined on Line 9 of Algorithm 1) is $O(L\tau^2)$-Lipschitz with respect to the norm $\|\cdot\|_{d,H}$ in $\mathcal{X}_{d,H,a_0,\overline{\alpha}}$.*

*Proof.* By $L$-Lipschitzness of $c_t$ with respect to $\|\cdot\|_1$, it suffices to show that for any $(p_1, M_1^{[1:H]}), (p_2, M_2^{[1:H]}) \in \mathcal{X}_{d,H,a_0,\overline{\alpha}}$, we have

$$\left\| x_t(p_1, M_1^{[1:H]}) - x_t(p_2, M_2^{[1:H]}) \right\|_1 \leqslant O(\tau^2) \left\| (p_1, M_1^{[1:H]}) - (p_2, M_2^{[1:H]}) \right\|_{d,H} \qquad (36)$$

$$\left\| u_t(p_1, M_1^{[1:H]}) - u_t(p_2, M_2^{[1:H]}) \right\|_1 \leqslant O(\tau^2) \left\| (p_1, M_1^{[1:H]}) - (p_2, M_2^{[1:H]}) \right\|_{d,H}. \qquad (37)$$

Fix $(p_1, M_1^{[1:H]}), (p_2, M_2^{[1:H]}) \in \mathcal{X}_{d,H,a_0,\overline{\alpha}}$, and write

$$\varepsilon := \max\left\{\|p_1 - p_2\|_1, \max_{j\in[H]}\left\|M_1^{[j]} - M_2^{[j]}\right\|_{1\to 1}\right\}.$$

Since $\varepsilon \leqslant \left\|(p_1, M_1^{[1:H]}) - (p_2, M_2^{[1:H]})\right\|_{d,H}$, it suffices to show that Eqs. (36) and (37) hold with $\varepsilon$ on the right-hand sides.

To verify Eq. (36) in this manner, we define, for $b \in \{1, 2\}$,

$$v_{i,b} := \lambda_{t-i,0}\bar{\lambda}_{t,i} \cdot B \cdot p_b + B\sum_{j=1}^{H}\lambda_{t,i+j} \cdot M_b^{[j]} \cdot w_{t-i-j} + \lambda_{t,i} \cdot w_{t-i}.$$

Since $\lambda_{t-i,0}\bar{\lambda}_{t,i} + \lambda_{t,i} + \sum_{j=1}^{H}\lambda_{t,i+j} \leqslant 1$, we have $\|v_{i,b}\|_1 \leqslant 1$ for each $i \in [t], b \in \{1,2\}$. Moreover, $\|v_{i,1} - v_{i,2}\|_1 \leqslant (\lambda_{t-i,0}\bar{\lambda}_{t,i} + \sum_{j=1}^{H}\lambda_{t,i+j}) \cdot \varepsilon \leqslant \varepsilon$. Write $\sigma_1 := \|p_1\|_1, \sigma_2 := \|p_2\|_1$, so that $|\sigma_1 - \sigma_2| \leqslant \varepsilon$ and $|(1-\sigma_1)^i - (1-\sigma_2)^i| \leqslant i\varepsilon$ for all $i \geqslant 1$. Also note that for each $b \in \{1,2\}$,

$$\sum_{i=1}^{t}(1-\sigma_b)^{i-1} \cdot \|v_{i,b}\|_1 = \sum_{i=1}^{t}(1-\sigma_b)^{i-1} \cdot \bar{\lambda}_{t,i-1} \cdot ((1-\gamma_{t-i}) \cdot \sigma_b + \gamma_{t-i})$$

$$= \sum_{i=1}^{t}(1-\sigma_b)^{i-1} \cdot \bar{\lambda}_{t,i-1} \cdot (1 - (1-\gamma_{t-i})(1-\sigma_b)) = 1, \qquad (38)$$

where the final equality follows since $\gamma_0 = 1$.

By Eq. (20), we have

$$x_t(p_1, M_1^{[1:H]}) - x_t(p_2, M_2^{[1:H]}) = \sum_{i=1}^{t}\left((1-\sigma_1)^{i-1}A^{i-1} \cdot v_{i,1} - (1-\sigma_2)^{i-1}A^{i-1} \cdot v_{i,2}\right).$$

We consider two cases, depending on the mixing time $\tau_A := t^{\mathsf{mix}}(A)$ of $A$:

**Case 1: $\tau_A \leqslant 4\tau$.** Let the stationary distribution of $A$ be denoted $p^\star \in \Delta^d$. Then

$$\left\|x_t(p_1, M_1^{[1:H]}) - x_t(p_2, M_2^{[1:H]})\right\|_1$$

$$= \left\|\sum_{i=1}^{t}\left((1-\sigma_1)^{i-1}A^{i-1}v_{i,1} - (1-\sigma_2)^{i-1}A^{i-1}v_{i,2}\right)\right\|_1$$

$$\leqslant \left\|\sum_{i=1}^{t}\left((1-\sigma_1)^{i-1}(A^{i-1}v_{i,1} - \|v_{i,1}\|_1 p^\star) - (1-\sigma_2)^{i-1}(A^{i-1}v_{i,2} - \|v_{i,2}\|_1 p^\star)\right)\right\|_1$$

$$\quad + \left\|\sum_{i=1}^{t}\left((1-\sigma_1)^{i-1}\|v_{i,1}\|_1 - (1-\sigma_2)^{i-1}\|v_{i,2}\|_1\right)p^\star\right\|_1$$

$$= \left\|\sum_{i=1}^{t}A^{i-1}\left((1-\sigma_1)^{i-1}v_{i,1} - (1-\sigma_2)^{i-1}v_{i,2}\right) - \left((1-\sigma_1)^{i-1}\|v_{i,1}\|_1 - (1-\sigma_2)^{i-1}\|v_{i,2}\|_1\right)p^\star\right\|_1$$

$$\leqslant \sum_{i=1}^{t}2^{1-\lfloor(i-1)/\tau_A\rfloor}\left\|(1-\sigma_1)^{i-1}v_{i,1} - (1-\sigma_2)^{i-1}v_{i,2} - \left((1-\sigma_1)^{i-1}\|v_{i,1}\|_1 - (1-\sigma_2)^{i-1}\|v_{i,2}\|_1\right)p^\star\right\|_1$$

$$\leqslant \sum_{i=1}^{t}2^{2-\lfloor(i-1)/\tau_A\rfloor}(i\varepsilon + \varepsilon)$$

$$\leqslant C\tau_A\varepsilon\sum_{i=0}^{\infty}\tau_A i 2^{-i}$$

$$\leqslant C'\tau^2\varepsilon$$

for some constants $C, C'$. Above, the second equality uses Eq. (38), and the second inequality uses Lemma 22 together with the fact that $A^{i-1}p^\star = p^\star$ and

$$\left\langle \mathbb{1}, \left((1-\sigma_1)^{i-1}v_{i,1} - (1-\sigma_2)^{i-1}v_{i,2}\right) - \left((1-\sigma_1)^{i-1}\|v_{i,1}\|_1 - (1-\sigma_2)^{i-1}\|v_{i,2}\|_1\right)\right\rangle = 0.$$

The final inequality uses the assumption that $\tau_A \leqslant 4\tau$.

**Case 2: $\tau_A > 4\tau$.** In this case, the assumption that $\mathcal{K}_\tau^\triangle(\mathcal{L})$ is nonempty together with the choice of $a_0$ in Line 1 of Algorithm 1 and Lemma 20 gives that $a_0 > 1/(96\tau)$. See Case 2 of the proof of Lemma 21 for more details of this argument, which uses the fact that $\tau_A > 96\tau$.

Since $(p_b, M_b^{[1:H]}) \in \mathcal{X}_{d,H,a_0,\overline{\alpha}}$ for $b \in \{1,2\}$, we have $\sigma_1, \sigma_2 \geqslant a_0 > 1/(96\tau)$. We may compute

$$\left\| x_t(p_1, M_1^{[1:H]}) - x_t(p_2, M_2^{[1:H]}) \right\|_1$$

$$= \left\| \sum_{i=1}^{t} \left((1-\sigma_1)^{i-1}A^{i-1}v_{i,1} - (1-\sigma_2)^{i-1}A^{i-1}v_{i,2}\right) \right\|_1$$

$$\leqslant \sum_{i=1}^{t} |(1-\sigma_1)^{i-1} - (1-\sigma_2)^{i-1}| + \sum_{i=1}^{t}(1-\sigma_1)^{i-1}\|v_{i,1} - v_{i,2}\|_1$$

$$\leqslant \sum_{i=2}^{t}\sum_{j=1}^{i-1}|\sigma_1 - \sigma_2|(1-\sigma_1)^{j-1}(1-\sigma_2)^{i-1-j} + \varepsilon\sum_{i=1}^{t}(1-\sigma_1)^{i-1}$$

$$\leqslant \sum_{i=2}^{t}(i-1)\varepsilon(1-1/(96\tau))^{i-2} + \varepsilon\sum_{i=1}^{t}(1-1/(96\tau))^{i-1}$$

$$\leqslant C\tau^2\varepsilon$$

for some constant $C$.

Thus, in both cases above, we have $\left\| x_t(p_1, M_1^{[1:H]}) - x_t(p_2, M_2^{[1:H]}) \right\|_1 \leqslant O(\tau\varepsilon)$, which verifies Eq. (36).

The proof of Eq. (37) is much simpler: we have

$$\left\| u_t(p_1, M_1^{[1:H]}) - u_t(p_2, M_2^{[1:H]}) \right\|_1 \leqslant \lambda_{t,0} \cdot \|p_1 - p_2\|_1 + \sum_{j=1}^{H}\lambda_{t,j} \cdot \left\| M_1^{[j]} - M_2^{[j]} \right\|_{1\to1} \leqslant \varepsilon,$$

since $\lambda_{t,0} + \cdots + \lambda_{t,H} = 1$. $\qquad\square$

*Proof of Theorem 7.* Set $\beta = \frac{\sqrt{2dH\ln d}}{\sqrt{T}}$, $\varepsilon = 1/T$, and $\mathcal{K}_\tau^\triangle := \mathcal{K}_\tau^\triangle(\mathcal{L})$. We will apply Corollary 15 to the sequence of iterates $(p_t, M_t^{[1:H]})$ produced in Algorithm 1, for the domain $\mathcal{X}_{d,H,a_0,\overline{\alpha}}$ (i.e., $a = a_0, b = \overline{\alpha}$). Note that Lemma 23 gives that $\ell_t$ is $O(L\tau^2)$-Lipschitz, for each $t \in [T]$. Thus Corollary 15 guarantees a regret bound (with respect to $\mathcal{X}_{d,H,a_0,\overline{\alpha}}$) of $O(L\tau^2\sqrt{dH\ln(d)T})$. Moreover, Eq. (19) of Corollary 15 ensures that the precondition Eq. (28) of Lemma 21 is satisfied. Thus, we may bound

$$\sum_{t=1}^{T}c_t(x_t, u_t) - \inf_{K\in\mathcal{K}_\tau^\triangle}\sum_{t=1}^{T}c_t(x_t(K), u_t(K))$$

$$\leqslant \sum_{t=1}^{T}\ell_t(p_t, M_t^{[1:H]}) - \inf_{K\in\mathcal{K}_\tau^\triangle}\sum_{t=1}^{T}c_t(x_t(K), u_t(K)) + O(T \cdot L\tau^3\log^3(1/\beta)\beta)$$

$$\leqslant \sum_{t=1}^{T}\ell_t(p_t, M_t^{[1:H]}) - \inf_{(p,M^{[1:H]})\in\mathcal{X}_{d,H,a_0,\overline{\alpha}}}\sum_{t=1}^{T}c_t(x_t(p, M^{[1:H]}), u_t(p, M^{[1:H]})) \qquad (39)$$

$$+ O(T \cdot L\tau^3\log^3(1/\beta)\beta) + \varepsilon$$

$$= \sum_{t=1}^{T}\ell_t(p_t, M_t^{[1:H]}) - \inf_{(p,M^{[1:H]})\in\mathcal{X}_{d,H,a_0,\overline{\alpha}}}\sum_{t=1}^{T}\ell_t(p, M^{[1:H]}) \qquad (40)$$

$$+ O(T \cdot L\tau^3 \log^3(1/\beta)\beta) + \varepsilon$$
$$\leqslant L\tau^2 \sqrt{dH \ln(d)T} + O(T \cdot L\tau^3 \log^3(1/\beta)\beta) + \varepsilon,$$

where the first inequality uses Lemma 21 together with Eq. (19) of Corollary 15, and the second inequality uses Lemma 17 with $\epsilon = 1/T$ (by the theorem assumption, the inequality $H \geqslant \tau\lceil \log_2(2LT^2/\epsilon) \rceil$ is indeed satisfied). Note that for the second inequality to hold, we also need that $\|K\|_{1\to 1} \geqslant a_0$ for all $K \in \mathcal{K}_\tau^\triangle$, which in particular requires (by Line 1) that $\|K\|_{1\to 1} \geqslant 1/(96\tau)$ if $t^{\mathsf{mix}}(A) > 4\tau$. But if $t^{\mathsf{mix}}(A) > 4\tau$, then for any $K \in \mathcal{K}_\tau^\triangle$ we have $t^{\mathsf{mix}}(A) > 4 \cdot t^{\mathsf{mix}}(\mathbb{A}_K)$ and hence $\|K\|_{1\to 1} \geqslant 1/(96\tau)$ by Lemma 20. Finally, the equality above uses the definition of $\ell_t$ in Algorithm 1, and the final inequality uses the regret bound of Corollary 15. By our choice of $\beta, \varepsilon$, we see that the overall policy regret is $\tilde{O}(L\tau^{7/2}d^{1/2}\sqrt{T})$, as desired. $\qquad\square$

# D  Proof of Lower Bounds

In this section, we formally state and prove the regret lower bounds Theorem 1 and Theorem 8. The former states that the comparator class for online control of standard LDSs cannot be broadened to all marginally stable (time-invariant, linear) policies; the latter states that the mixing time assumption cannot be removed from the comparator class for online control of simplex LDSs. Both results hold even in constant dimension.

The basic idea is the same for both proofs: we construct two systems $\mathcal{L}^0, \mathcal{L}^1$ which are identical until time $T/2$, but then at time $T/2$ experience differing perturbations of constant magnitude. The costs are zero until time $T/2$, after which they penalize distance to a prescribed state (and can in fact be taken to be the same for both systems). The optimal strategy in the first $T/2$ time steps therefore depends on which system the controller is in, but the controller does not observe this until time $T/2$, and hence will necessarily incur regret with respect to the optimal policy.

Formalizing this intuition requires two additional pieces: first, for both systems there must be a near-optimal time-invariant linear policy. This can be achieved by careful design of the dynamics, perturbations, and costs. Second, if the controller finds itself in a high-cost state at time $T/2 + 1$, it must be unable to reach a low-cost state without incurring $\Omega(T)$ total cost along the way. In the standard LDS setting, we achieve this by setting the transition matrices $A, B$ so that $\|B\| = O(1/T)$ (i.e. so constant-size controls have small effect on the state) and adding a penalty of $|u_t|$ to the cost for $t > T/2$. In the simplex LDS setting, we achieve this by our choice of the valid constraint set $\mathcal{I}$ (which enforces that $\|u_t\|_1 = O(1/T)$ for all $t$).

See Fig. 3 for a pictorial explanation of the proof in the simplex LDS setting.

## D.1  Proof of Theorem 1

In this section we give a formal statement and proof of Theorem 1. Recall the definition of an LDS (Definition 9). We define the class $\mathcal{K}_\kappa(\mathcal{L})$ of policies that $\kappa$-*marginally stabilize* $\mathcal{L}$ below; it is equivalent to the class $\mathcal{K}_{\kappa,\rho}(\mathcal{L})$ of policies that $(\kappa, \rho)$-strongly stabilize $\mathcal{L}$ (Definition 4) with $\rho = 0$.

**Definition 24** (Marginal stabilization). *A matrix $M \in \mathbb{R}^{d \times d}$ is $\kappa$-marginally stable if there is a matrix $H \in \mathbb{R}^{d \times d}$ so that $\|H^{-1}MH\| \leqslant 1$ and $\|M\|, \|H\|, \|H^{-1}\| \leqslant \kappa$. A matrix $K \in \mathbb{R}^{d \times d}$ is said to $\kappa$-marginally stabilize an LDS with transition matrices $A, B \in \mathbb{R}^{d \times d}$ if $A + BK$ is $\kappa$-marginally stable. For $\kappa > 0$ and an LDS $\mathcal{L}$ on $\mathbb{R}^d$, we define $\mathcal{K}_\kappa(\mathcal{L})$ to be the set of linear, time-invariant policies $x \mapsto Kx$ where $K \in \mathbb{R}^{d \times d}$ $\kappa$-marginally stabilizes $\mathcal{L}$.*

We also introduce a standard regularity assumption on cost functions:[15]

**Assumption 2.** *Let $L > 0$. We say that cost functions $(c_t)_t$, where $c_t : \mathbb{R}^{d_x} \times \mathbb{R}^{d_u} \to \mathbb{R}$, are $L$-regular if $c_t$ is convex and $L$-Lipschitz with respect to the Euclidean norm for all $t$.*

**Theorem 25** (Formal statement of Theorem 1). *Let* `Alg` *be any randomized algorithm for online control with the following guarantee:*

---

[15]Technically, in this setting of general LDSs where the state domain is unbounded, Assumption 1 is stronger than the assumption on cost functions made in prior work on non-stochastic control [1], because it enforces a uniform Lipschitzness bound on the entire domain. But we are proving a *lower bound* in this section, so this strengthening only makes our result stronger.

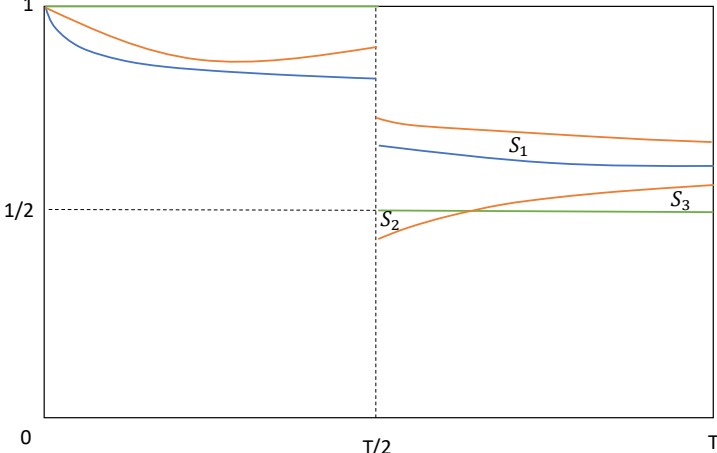

**Figure 3:** An intuitive illustration of $x_t(2)$ in the lower bound for simplex LDS (Theorem 30). The blue curve is the trajectory of $\pi^0$, the "decreasing" comparator policy, in the system $\mathcal{L}^0$, which has the smaller perturbation. The green curve is $\pi^1$, the "lazy" comparator policy, in the system $\mathcal{L}^1$, which has the larger perturbation. The orange curves correspond to the trajectories of an arbitrary policy $\pi$ under the two different perturbation sequences. The sum of regret under the two perturbation sequences is equal to the area $S_1 + S_2 + S_3$, which is shown to be $\Omega(T)$ for any $h$.

*Let $d, T \in \mathbb{N}$ and $\kappa > 0$, and let $\mathcal{L} = (A, B, x_1, (w_t)_t, (c_t)_t)$ be an LDS with state space and control space $\mathbb{R}^d$; $L$-regular cost functions $(c_t)_t$ (Assumption 2); and perturbations $(w_t)_t$ satisfying $\|w_t\|_2 \leqslant L$ for all $t$. Then the iterates $(x_t, u_t)_{t=1}^T$ produced by Alg with input $(A, B, \kappa, T)$ on interaction with $\mathcal{L}$ satisfy*

$$\mathbf{regret}_{\mathcal{K}_\kappa(\mathcal{L})} := \mathbb{E}\left[\sum_{t=1}^T c_t(x_t, u_t)\right] - \inf_{K \in \mathcal{K}_\kappa(\mathcal{L})} \sum_{t=1}^T c_t(x_t^{\mathcal{L},K}, u_t^{\mathcal{L},K}) \leqslant f(d, \kappa, L, T) \quad (41)$$

*where $(x_t^{\mathcal{L},K}, u_t^{\mathcal{L},K})_{t=1}^T$ are the iterates produced by following policy $x \mapsto Kx$ in system $\mathcal{L}$ for all $t \in [T]$.*

*Then $f(1, 1, 1, T) = \Omega(T)$.*

**Remark 26.** In the above theorem statement, if $\mathcal{K}_\kappa(\mathcal{L})$ were replaced with $\mathcal{K}_{\kappa,\rho}(\mathcal{L})$, the class of linear time-invariant policies that $(\kappa, \rho)$-*strongly stabilize* $\mathcal{L}$, then the main result of [1] would imply that in fact there is a (deterministic) algorithm GPC with regret at most $\mathrm{poly}(d, \kappa, L, \rho^{-1}) \cdot \sqrt{T} \log(T)$ on any LDS $\mathcal{L}$ satisfying the above conditions. Thus, Theorem 25 indeed provides a converse to [1].

We prove Theorem 25 by constructing a simple distribution over LDSs on which any algorithm must incur $\Omega(T)$ regret in expectation. Let $\beta \geqslant 2$ be a constant that we will determine later, and fix $T \geqslant \beta$. Recall that we denote an LDS on $\mathbb{R}^d$ using the notation $\mathcal{L} = (A, B, x_1, (w_t)_t, (c_t)_t)$, where $A, B \in \mathbb{R}^{d \times d}$. We define two LDSs on $\mathbb{R}$ as follows:

$$\mathcal{L}^0 := (1, -\beta/T, x_1, (w_t^0)_t, (c_t)_t),$$
$$\mathcal{L}^1 := (1, -\beta/T, x_1, (w_t^1)_t, (c_t)_t),$$

where the (common) initial state is $x_1 = 1$, the (common) cost functions $(c_t)_t$ are defined as

$$c_t(x, u) := \begin{cases} |x| + |u| & \text{if } t > T/2 \\ 0 & \text{otherwise} \end{cases},$$

the perturbations of $\mathcal{L}^0$ are $w_t^0 := 0$ for all $t$, and the perturbations of $\mathcal{L}^1$ are

$$w_t^1 := \begin{cases} -1 & \text{if } t = T/2 \\ 0 & \text{otherwise} \end{cases}.$$

For simplicity, we assume that $T/2$ is an integer. Thus, at all times $t \neq T/2$, the two systems have identical dynamics

$$x_{t+1} := x_t - \frac{\beta}{T} u_t,$$

but at time $t = T/2$, system $\mathcal{L}^1$ experiences a negative perturbation of magnitude 1, whereas $\mathcal{L}^0$ does not. The following lemma characterizes the performance of two time-invariant linear policies $\pi^0, \pi^1$ for $\mathcal{L}^0, \mathcal{L}^1$ respectively:

**Lemma 27.** *Define $\pi^0, \pi^1 : \mathbb{R} \to \mathbb{R}$ by $\pi^0(x) = x$ and $\pi^1(x) = 0$. Then:*

- *Policy $\pi^0$ is an element of $\mathcal{K}_1(\mathcal{L}^0)$, and the iterates $(x_t^{\mathcal{L}^0,\pi^0}, u_t^{\mathcal{L}^0,\pi^0})_{t=1}^T$ produced by following $\pi^0$ in system $\mathcal{L}^0$ satisfy*

$$\sum_{t=1}^T c_t(x_t^{\mathcal{L}^0,\pi^0}, u_t^{\mathcal{L}^0,\pi^0}) \leq \frac{2T}{\beta} e^{-\beta/2}.$$

- *Policy $\pi^1$ is an element of $\mathcal{K}_1(\mathcal{L}^1)$, and the iterates $(x_t^{\mathcal{L}^1,\pi^1}, u_t^{\mathcal{L}^1,\pi^1})_{t=1}^T$ produced by following $\pi^1$ in system $\mathcal{L}^1$ satisfy*

$$\sum_{t=1}^T c_t(x_t^{\mathcal{L}^1,\pi^1}, u_t^{\mathcal{L}^1,\pi^1}) = 0.$$

*Proof.* Note that $\pi^0, \pi^1$ are both time-invariant linear policies. The inclusion $\pi^0 \in \mathcal{K}_1(\mathcal{L}^0)$ is immediate from the fact that $\mathcal{L}^0$ has transitions $A = 1, B = -\beta/T$, and $|A + B| \leq 1$. Similarly, $\pi^1 \in \mathcal{K}_1(\mathcal{L}^1)$ because $|A| \leq 1$. To bound the total cost of $\pi^0$ on $\mathcal{L}^0$, note that $u_t^{\mathcal{L}^0,\pi^0} = x_t^{\mathcal{L}^0,\pi^0} = (1 - \beta/T)^{t-1}$ for all $t \in [T]$. Hence,

$$\sum_{t=1}^T c_t(x_t^{\mathcal{L}^0,\pi^0}, u_t^{\mathcal{L}^0,\pi^0}) = 2 \sum_{t=T/2+1}^T \left(1 - \frac{\beta}{T}\right)^{t-1} \leq \frac{2T}{\beta}\left(1 - \frac{\beta}{T}\right)^{T/2} \leq \frac{2T}{\beta} e^{-\beta/2}.$$

Moreover, we have $x_t^{\mathcal{L}^1,\pi^1} = \mathbb{1}[t \leq T/2]$ and $u_t^{\mathcal{L}^1,\pi^1} = 0$ for all $t \in [T]$, from which it is clear that $\sum_{t=1}^T c_t(x_t^{\mathcal{L}^1,\pi^1}, u_t^{\mathcal{L}^1,\pi^1}) = 0$. $\qquad\square$

Next, we show that the total cost of any trajectory $(x_t, u_t)_{t=1}^T$ can be lower bounded in terms of $|x_{T/2+1}|$ in both $\mathcal{L}^0$ and $\mathcal{L}^1$:

**Lemma 28.** *Let* Alg *be any randomized algorithm for online control, and let $b \in \{0, 1\}$. The (random) trajectory $(x_t, u_t)_{t=1}^T$ produced by* Alg *in system $\mathcal{L}^b$ satisfies the inequality*

$$\sum_{t=1}^T c_t(x_t, u_t) = \sum_{t=T/2+1}^T |x_t| + |u_t| \geq \frac{T}{2\beta}|x_{T/2+1}|$$

*with probability 1.*

*Proof.* By definition of $\mathcal{L}^0, \mathcal{L}^1$, any valid trajectory in $\mathcal{L}^b$ satisfies $|u_t| = \frac{T}{\beta}|x_{t+1} - x_t|$ for all $T/2 < t < T$. We consider two cases:

1. If $\min_{T/2 < t \leq T} |x_t| \geq \frac{1}{2}|x_{T/2+1}|$, then

$$\sum_{t=T/2+1}^T |x_t| + |u_t| \geq \frac{T}{4}|x_{T/2+1}| \geq \frac{T}{2\beta}|x_{T/2+1}|$$

since $\beta \geq 2$.

2. If $\min_{T/2 < t \leqslant T} |x_t| < \frac{1}{2}|x_{T/2+1}|$, then

$$\sum_{t=T/2+1}^{T} |x_t| + |u_t| \geqslant \frac{T}{\beta} \sum_{t=T/2+1}^{T-1} |x_{t+1} - x_t| \geqslant \frac{T}{\beta} \left| |x_{T/2+1}| - \min_{T/2 < t \leqslant T} |x_t| \right| \geqslant \frac{T}{2\beta}|x_{T/2+1}|$$

by the triangle inequality.

In both cases the claimed inequality holds. $\qquad\square$

We can now prove Theorem 25.

*Proof of Theorem 25.* Let $b \sim \mathrm{Unif}(\{0,1\})$ be an unbiased random bit, and let $(x_t, u_t)_{t=1}^{T}$ be the (random) trajectory produced by executing Alg on $\mathcal{L}^b$. On the one hand, by Eq. (41) applied to $\mathcal{L}^0$ and $\mathcal{L}^1$, we have

$$\mathbb{E}\left[ \sum_{t=1}^{T} c_t(x_t, u_t) \right]$$

$$\leqslant f(1,1,1,T) + \frac{1}{2}\left( \inf_{K \in \mathcal{K}_1(\mathcal{L}^0)} \sum_{t=1}^{T} c_t(x_t^{\mathcal{L}^0, K}, u_t^{\mathcal{L}^0, K}) + \inf_{K \in \mathcal{K}_1(\mathcal{L}^1)} \sum_{t=1}^{T} c_t(x_t^{\mathcal{L}^1, K}, u_t^{\mathcal{L}^1, K}) \right)$$

$$\leqslant f(1,1,1,T) + \frac{T}{\beta} e^{-\beta/2} \qquad (42)$$

where the first inequality uses the fact that the cost functions $(c_t)_t$ are convex and 1-Lipschitz and that $|w_t^0|, |w_t^1| \leqslant 1$ for all $t \in [T]$; and the second inequality is by Lemma 27. On the other hand, by Lemma 28, we have

$$\mathbb{E}\left[ \sum_{t=1}^{T} c_t(x_t, u_t) \right] \geqslant \frac{T}{2\beta} \mathbb{E}[|x_{T/2+1}|]$$

$$= \frac{T}{2\beta} \mathbb{E}\left[ \left| x_{T/2} - \frac{\beta}{T} u_{T/2} - b \right| \right]$$

$$\overset{(\star)}{=} \frac{T}{2\beta}\left( \frac{1}{2}\mathbb{E}\left[ \left| x_{T/2} - \frac{\beta}{T} u_{T/2} \right| \right] + \frac{1}{2}\mathbb{E}\left[ \left| x_{T/2} - \frac{\beta}{T} u_{T/2} - 1 \right| \right] \right)$$

$$\geqslant \frac{T}{4\beta}\left( \mathbb{E}\left[ x_{T/2} - \frac{\beta}{T} u_{T/2} \right] + \left| \mathbb{E}\left[ x_{T/2} - \frac{\beta}{T} u_{T/2} \right] - 1 \right| \right) \geqslant \frac{T}{4\beta}, \qquad (43)$$

where the key equality $(\star)$ uses the fact that $\mathcal{L}^0, \mathcal{L}^1$ are identical up until and including time $T/2$, and hence $(x_{T/2}, u_{T/2})$ is independent of $b$. Comparing Eq. (43) with Eq. (42) yields that

$$f(1,1,1,T) \geqslant \frac{T}{4\beta} - \frac{T}{\beta} e^{-\beta/2} = \Omega(T)$$

for any sufficiently large constant $\beta$. $\qquad\square$

### D.2 Proof of Theorem 8

**Definition 29.** Let $0 \leqslant \underline{\alpha} \leqslant \overline{\alpha} \leqslant 1$ and let $\mathcal{I} := \bigcup_{\alpha \in [\underline{\alpha}, \overline{\alpha}]} \Delta_\alpha^d$. We define $\mathcal{K}(\mathcal{I})$ to be the set of linear, time-invariant policies $x \mapsto Kx$ where $K \in \bigcup_{\alpha \in [\underline{\alpha}, \overline{\alpha}]} \mathbb{S}_\alpha^d$.

**Theorem 30** (Formal statement of Theorem 8)**.** *Let* Alg *be any randomized algorithm for online control with the following guarantee:*

*Let $d, T \in \mathbb{N}$ and $\mathcal{I} := \bigcup_{\alpha \in [0, \overline{\alpha}]} \Delta_\alpha^d$ for some $\overline{\alpha} \in (0,1)$. Let $\mathcal{L} = (A, B, \mathcal{I}, x_1, (\gamma_t)_t, (w_t)_t, (c_t)_t)$ be a simplex LDS with state space $\Delta^d$ and cost functions $(c_t)_t$ satisfying Assumption 1 with Lipschitz parameter $L > 0$. Then the iterates $(x_t, u_t)_{t=1}^{T}$ produced by* Alg *with input $(A, B, \mathcal{I}, T)$ on interaction with $\mathcal{L}$ satisfy*

$$\mathbf{regret}_{\mathcal{K}(\mathcal{I})} := \mathbb{E}\left[ \sum_{t=1}^{T} c_t(x_t, u_t) \right] - \inf_{K \in \mathcal{K}(\mathcal{I})} \sum_{t=1}^{T} c_t(x_t^{\mathcal{L}, K}, u_t^{\mathcal{L}, K}) \leqslant f(d, L, \overline{\alpha}, T) \qquad (44)$$

*where $(x_t^{\mathcal{L},K}, u_t^{\mathcal{L},K})_{t=1}^T$ are the iterates produced by following policy $x \mapsto Kx$ in system $\mathcal{L}$ for all $t \in [T]$.*

*For any sufficiently large constant $\beta$, if we define $\overline{\alpha}(T) := \beta/T$, then $f(1, 1, \overline{\alpha}(T), T) = \Omega(T)$.*

We define two simplex LDSs on $\Delta^2$ as follows:

$$\mathcal{L}^0 := (I_2, I_2, \mathcal{I}, x_1, (\gamma_t)_t, (w_t^0)_t, (c_t)_t)$$
$$\mathcal{L}^1 := (I_2, I_2, \mathcal{I}, x_1, (\gamma_t)_t, (w_t^1)_t, (c_t)_t)$$

where $I_2 \in \mathbb{R}^{2\times 2}$ is the identity matrix, the (common) valid control set is $\mathcal{I} := \bigcup_{\alpha \in [0, \beta/T]} \Delta_\alpha^d$, the (common) initial state is $x_1 = (0, 1)$, the (common) cost functions $(c_t)_t$ are defined as

$$c_t(x, u) := \begin{cases} |x(2) - 1/2| & \text{if } t > T/2 \\ 0 & \text{otherwise} \end{cases},$$

the (common) perturbation strengths are $\gamma_t := \frac{1}{2}\mathbb{1}[t = T/2]$, and the perturbations of $\mathcal{L}^0$ are $w_t^0 := (1/2, 1/2)$ for all $t$ whereas the perturbations of $\mathcal{L}^1$ are $w_t^1 := (1, 0)$ for all $t$. Thus, for both systems, the dynamics are described by

$$x_{t+1} := (1 - \|u_t\|_1)x_t + u_t$$

for all $t \neq T/2$.

**Lemma 31.** *Define $\pi^0, \pi^1 : \Delta^2 \to \bigcup_{\alpha \in [0,1]} \Delta^d$ by $\pi^0(x) := \frac{\beta}{T}(1/2, 1/2)$ and $\pi^1(x) := (0, 0)$. Then $\pi^0, \pi^1 \in \mathcal{K}(\mathcal{I})$, and:*

- *The iterates $(x_t^{\mathcal{L}^0, \pi^0}, u_t^{\mathcal{L}^0, \pi^0})_{t=1}^T$ produced by following $\pi^0$ in system $\mathcal{L}^0$ satisfy*

$$\sum_{t=1}^T c_t(x_t^{\mathcal{L}^0, \pi^0}, u_t^{\mathcal{L}^0, \pi^0}) \leq \frac{T}{\beta}e^{-\beta/2}.$$

- *The iterates $(x_t^{\mathcal{L}^1, \pi^1}, u_t^{\mathcal{L}^1, \pi^1})_{t=1}^T$ produced by following $\pi^1$ in system $\mathcal{L}^1$ satisfy*

$$\sum_{t=1}^T c_t(x_t^{\mathcal{L}^1, \pi^1}, u_t^{\mathcal{L}^1, \pi^1}) = 0.$$

*Proof.* The fact that $\pi^0, \pi^1 \in \mathcal{K}(\mathcal{I})$ is immediate from Definition 29 and the choice of $\mathcal{I}$. To bound the total cost of $\pi^0$ on $\mathcal{L}^0$, note that $x_{t+1}^{\mathcal{L}^0, \pi^0}(2) - 1/2 = (1 - \beta/T)(x_t(2) - 1/2)$ for all $t \neq T/2$, and $x_{t+1}^{\mathcal{L}^0, \pi^0}(2) - 1/2 = (1/2)(1 - \beta/T)(x_t(2) - 1/2)$ for $t = T/2$. Thus,

$$\sum_{t=1}^T c_t(x_t^{\mathcal{L}^0, \pi^0}, u_t^{\mathcal{L}^0, \pi^0}) = \sum_{t=T/2+1}^T |x_t^{\mathcal{L}^0, \pi^0}(2) - 1/2| \leq \sum_{t=T/2+1}^T (1 - \beta/T)^{t-1} \leq \frac{T}{\beta}e^{-\beta/2}.$$

Moreover, we have $x_t^{\mathcal{L}^1, \pi^1} = (0, 1)$ for all $t \leq T/2$ and $x_t^{\mathcal{L}^1, \pi^1} = (1/2, 1/2)$ for all $t > T/2$, so indeed $\sum_{t=1}^T c_t(x_t^{\mathcal{L}^1, \pi^1}, u_t^{\mathcal{L}^1, \pi^1}) = 0$ as claimed. $\square$

**Lemma 32.** *Let* `Alg` *be any randomized algorithm for online control, and let $b \in \{0, 1\}$. The (random) trajectory $(x_t, u_t)_{t=1}^T$ produced by* `Alg` *in system $\mathcal{L}^b$ satisfies the inequality*

$$\sum_{t=1}^T c_t(x_t, u_t) = \sum_{t=T/2+1}^T |x_t(2) - 1/2| \geq \frac{T}{8\beta}|x_{T/2+1}(2) - 1/2|^2 - 1.$$

*Proof.* By definition of $\mathcal{L}^0, \mathcal{L}^1$ and the valid control set $\mathcal{I}$, any valid trajectory in $\mathcal{L}^b$ satisfies $\|x_t - x_{t+1}\|_1 \leq 2\|u_t\|_1 \leq 2\beta/T$ for all $T/2 < t < T$. It follows that $|x_{t+1}(2) - 1/2| \geq |x_t(2) - 1/2| - 2\beta/T$ for all such $t$, and hence

$$\sum_{t=T/2+1}^T |x_t(2) - 1/2| \geq \sum_{n=1}^{T/2} \max\left(0, |x_{T/2+1}(2) - 1/2| - \frac{2\beta n}{T}\right)$$

$$\geqslant \frac{|x_{T/2+1}(2) - 1/2|}{2} \cdot \left\lfloor \frac{T}{4\beta} |x_{T/2+1}(2) - 1/2| \right\rfloor$$

$$\geqslant \frac{T}{8\beta} |x_{T/2+1}(2) - 1/2|^2 - 1$$

as claimed. $\qquad\square$

*Proof of Theorem 30.* Let $b \sim \mathrm{Unif}(\{0,1\})$ be an unbiased random bit, and let $(x_t, u_t)_{t=1}^T$ be the (random) trajectory produced by executing `Alg` on $\mathcal{L}^b$. On the one hand, by Eq. (44) applied to $\mathcal{L}^0$ and $\mathcal{L}^1$, we have

$$\mathbb{E}\left[\sum_{t=1}^T c_t(x_t, u_t)\right]$$

$$\leqslant f(1,1,\beta/T,T) + \frac{1}{2}\left(\inf_{K \in \mathcal{K}(\mathcal{I})} \sum_{t=1}^T c_t(x_t^{\mathcal{L}^0,K}, u_t^{\mathcal{L}^0,K}) + \inf_{K \in \mathcal{K}(\mathcal{I})} \sum_{t=1}^T c_t(x_t^{\mathcal{L}^1,K}, u_t^{\mathcal{L}^1,K})\right)$$

$$\leqslant f(1,1,\beta/T,T) + \frac{T}{2\beta} e^{-\beta/2} \tag{45}$$

where the first inequality uses the definition of $\mathcal{I}$ and the fact that the cost functions $(c_t)_t$ are convex and 1-Lipschitz per Assumption 1; and the second inequality is by Lemma 31. On the other hand, by Lemma 32, we have

$$1 + \mathbb{E}\left[\sum_{t=1}^T c_t(x_t, u_t)\right]$$

$$\geqslant \frac{T}{8\beta}\mathbb{E}[(x_{T/2+1}(2) - 1/2)^2]$$

$$= \frac{T}{8\beta}\mathbb{E}\left[\left(\frac{(1 - \|u_{T/2}\|_1)x_{T/2}(2) + u_{T/2}(2)}{2} - \frac{1+b}{4}\right)^2\right]$$

$$\overset{(\star)}{=} \frac{T}{8\beta}\left(\frac{1}{2}\mathbb{E}\left[\left(\frac{(1 - \|u_{T/2}\|_1)x_{T/2}(2) + u_{T/2}(2)}{2} - \frac{1}{4}\right)^2\right] + \frac{1}{2}\mathbb{E}\left[\left(\frac{(1 - \|u_{T/2}\|_1)x_{T/2}(2) + u_{T/2}(2)}{2} - \frac{1}{2}\right)^2\right]\right)$$

$$\geqslant \frac{T}{16\beta}\left(\left(\mathbb{E}\left[\frac{(1 - \|u_{T/2}\|_1)x_{T/2}(2) + u_{T/2}(2)}{2}\right] - \frac{1}{4}\right)^2 + \left(\mathbb{E}\left[\frac{(1 - \|u_{T/2}\|_1)x_{T/2}(2) + u_{T/2}(2)}{2}\right] - \frac{1}{2}\right)^2\right)$$

$$\geqslant \frac{T}{1024\beta}. \tag{46}$$

where the key equality $(\star)$ uses the fact that $\mathcal{L}^0, \mathcal{L}^1$ are identical up until and including time $T/2$, and hence $(x_{T/2}, u_{T/2})$ is independent of $b$. Comparing Eq. (46) with Eq. (45) yields that

$$f(1,1,\beta/T,T) \geqslant \frac{T}{1024\beta} - 1 - \frac{T}{2\beta} e^{-\beta/2} = \Omega(T)$$

for any sufficiently large constant $\beta$. $\qquad\square$

# E Implementation details

In this section we describe the version of `GPC-Simplex` (Algorithm 1) implemented for our experiments. First, the dynamical systems in our experiments are non-linear. The `GPC-Simplex` algorithm is still practical and applicable in such settings – concretely, any setting with update rule Eq. (9) – but of course several modifications/generalizations must be made:

1. The algorithm takes as input the function $f$ describing the dynamics in Eq. (9), rather than transition matrices $A, B$. Accordingly, in Line 8, the expression $(1 - \|u_t\|_1)Ax_t + Bu_t$

(which exactly corresponds to the noiseless update rule in a simplex LDS) is replaced by $f(x_t, u_t)$. Moreover, in Line 9, the hypothetical iterates $x_t(p, M^{[1:H]}), u_t(p, M^{[1:H]})$ under policy $\pi^{p, M^{[1:H]}}$ are computed using the update rule $f$.

2. The algorithm directly takes as input a learning rate $\eta$ for the mirror descent subroutine, rather than the mixing time bound $\tau$. In our experiments, we always set $\eta := \sqrt{dH \ln(H)}/(2\sqrt{T})$.

3. We always parametrize our systems so that the valid control set is the space of distributions $\Delta^d$. Hence, the domain used for mirror descent is $\mathcal{X}_{d,H,1,1}$. Mirror descent is implemented by exponential weights updates with learning rate $\eta$ and uniform initialization.

We remark that the above (natural) modifications to `GPC-Simplex` are analogous to the modifications to `GPC` made by [2] to perform online control for nonlinear systems.

## F  Experiments: Controlled SIR model

In this section, we provide additional experiments in the controlled SIR model. Specifically, in Appendix F.1 we provide experimental evaluations when there are perturbations to the system (i.e. $\gamma_t$ is not always 0 in Eq. (9)). In Appendix F.2 we vary the parameters of the SIR model.

### F.1  Control in presence of perturbations

We experiment with the SIR system Eq. (11) with the following parameters:
$$\beta = 0.5, \quad \theta = 0.03, \quad \xi = 0.005,$$
and cost function given by:
$$c_t(x_t, u_t) = c_3 \cdot x_t(2)^2 + c_2 \cdot x_t(1)u_t(1).$$

We test the performance of our algorithm on $(c_2, c_3) = (1, 5)$. In addition, we add a perturbation sequence $w_t = [0, 1, 0], \forall 1 \leqslant t \leqslant 200$. $\gamma_t \sim 0.01 \cdot \text{Ber}(0.2), \forall 1 \leqslant t \leqslant 200$.

Fig. 4 shows comparison of the costs over $T = 200$ time steps incurred by `GPC-simplex` to that of always executing $u_t = [1, 0]$ (full prevention) and that of always executing $u_t = [0, 1]$ (no prevention). In addition to cost, we plot the value of $u_t(2)$ over time, representing how relaxed prevention measure evolves over time according to `GPC-simplex`.

### F.2  Alternative parameter settings

We experiment with two SIR systems with different set of parameters. The first uses the following parameters:
$$\beta = 0.5, \quad \theta = 0.03, \quad \xi = 0.005,$$
whereas the second uses the following parameters:
$$\beta = 0.3, \quad \theta = 0.05, \quad \xi = 0.001.$$
In both cases, the cost function is:
$$c_t(x_t, u_t) = c_3 \cdot x_t(2)^2 + c_2 \cdot x_t(1)u_t(1).$$

For both experiments, we test the performance of our algorithm on different choices of parameters for the cost function. In particular, we test the parameter tuples:
$$(c_2, c_3) \in \{(1, 20), (1, 10), (1, 5), (1, 1)\}.$$
Figs. 5 and 6 show comparison of the costs over $T = 200$ time steps incurred by `GPC-Simplex` to that of always executing $u_t = [1, 0]$ (full prevention) and that of always executing $u_t = [0, 1]$ (no prevention). Specifically, Fig. 5 uses the first set of parameters above, and Fig. 6 uses the second set. In addition to cost, we plot the value of $u_t(2)$ over time, representing how the effective transmission rate evolves over time according to `GPC-Simplex`.

We notice that our algorithm consistently outperforms the two baselines. No matter how we set the parameters, our algorithm will outperform the full-intervention baseline since its cumulative cost grows linearly with time. As $c_3$ gets larger, the gap between our algorithm and the no-intervention baseline becomes smaller, since the optimal policy with a high cost on control is basically playing no control.

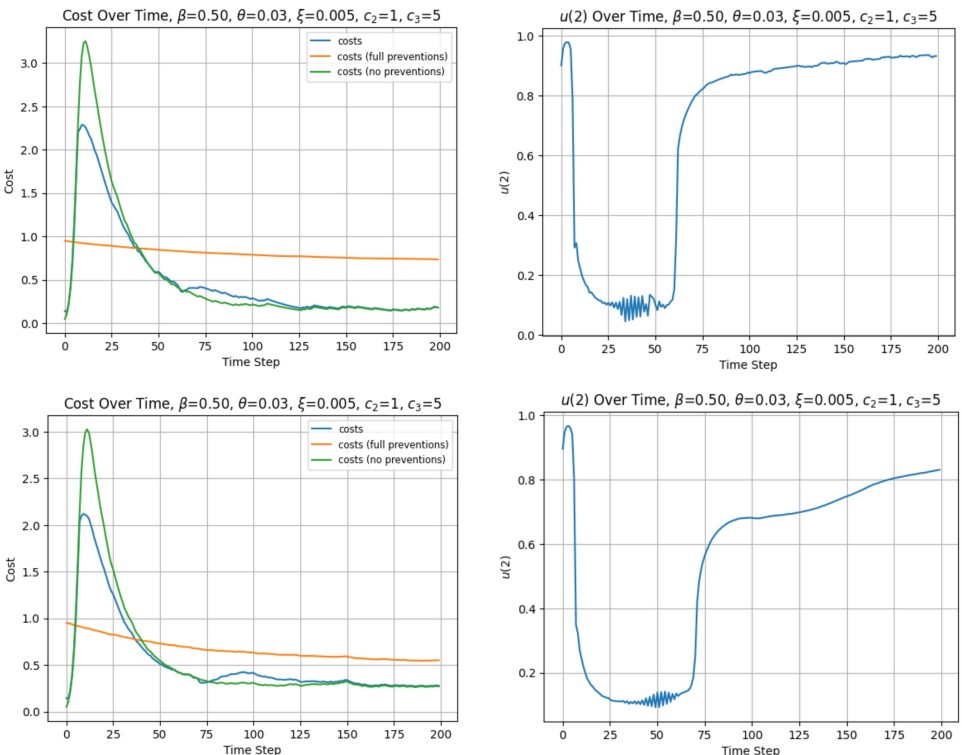

**Figure 4:** SIR with perturbations. $T = 200$. Initial state $x_1 = [0.9, 0.1, 0]$. `GPC-Simplex` parameter $H = 5$. **Top**: Perturbation sequence: $w_t = [0, 1, 0], \forall 1 \leqslant t \leqslant 200$. $\gamma_t \sim 0.01 \cdot \mathrm{Ber}(0.2), \forall 1 \leqslant t \leqslant 200$. **Bottom**: Perturbation sequence: $\forall t, w_t$ is a normalized uniform random vector. $\gamma_t = 0.01, \forall 1 \leqslant t \leqslant 200$.

# G  Experiments: Controlling hospital flows

In this section, we provide more details regarding the setup and experiments in Section 4.1.

The continuous time dynamical system considered by [27] is the following: let $S(t), I(t)$ denote the susceptible and infected fraction of the population at time $t$, and let $\sigma(t)$ denote the control at time $t$. The system has some initial state $(S(0), I(0))$ in the set

$$\mathcal{D} := \{(x_0, y_0) : x_0 > 0, y_0 > 0, x_0 + y_0 \leqslant 1\},$$

reflecting the constraint that $S(0), I(0)$ represent disjoint proportions of a population, and the system evolves according to the differential equation

$$S'(t) = -\gamma \sigma(t) I(t) S(t), \tag{47}$$

$$I'(t) = \gamma \sigma(t) I(t) S(t) - \gamma I(t). \tag{48}$$

where $\gamma > 0$ is some fixed model parameter, and the control $\sigma(t)$ models a non-pharmaceutical intervention (NPI) inducing a time-dependent reproduction number $\sigma(t) \in [0, \sigma_0]$, where $\sigma_0$ is the base reproduction number in the absence of interventions. In most examples in [27], including the example of controlling hospital flows, the parameter settings $\sigma_0 = 3$ and $\gamma = 0.1$ are used. This means that the natural discretization of Eq. (48) is in fact equivalent to Eq. (11) with transmission rate $\beta := \gamma \sigma_0 = 0.3$, recovery rate $\theta := \gamma = 0.1$, loss-of-immunity rate $\xi := 0$, no perturbations (i.e. $\gamma_t = 0$ for all $t$), and control

$$u_t := \left(1 - \frac{\sigma(t)}{\sigma_0}, \frac{\sigma(t)}{\sigma_0}\right),$$

at each time $t$.

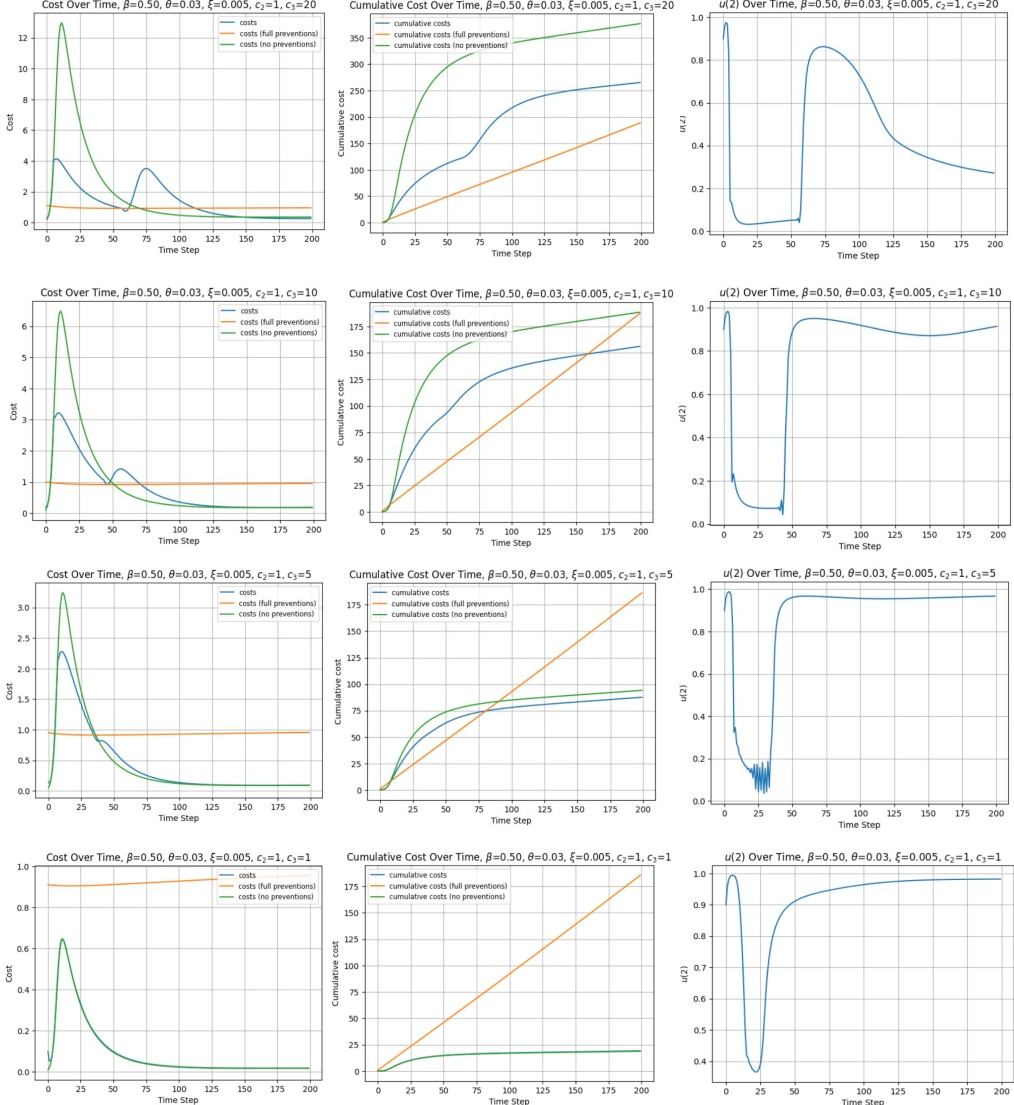

**Figure 5:** Control with costs: control over $T = 200$ steps. $\gamma_t = 0, \forall t$. SIR parameters: $\beta = 0.5, \theta = 0.03, \xi = 0.005$. Initial state $x_1 = [0.9, 0.1, 0]$. `GPC-Simplex` parameters: $H = 5$. **Left**: instantaneous cost over time, compared with that of no control (green) and full control (orange). **Middle**: cumulative cost over time. **Right**: $u_t(2)$ output by `GPC-Simplex` over time. $(c_2, c_3)$ values (from top to bottom rows): $(1, 20), (1, 10), (1, 5), (1, 1)$.

The goal in [27] is the following: given an initial state $(S(0), I(0))$ along with a horizon length $T > 0$ and the parameters listed above, choose an admissible control function $\sigma : [0, T] \to [0, \sigma_0]$ to minimize the loss

$$J := -S_\infty(S(T), I(T), \sigma_0) + \int_0^T L(S(t), I(t), \sigma(t))dt,$$

where $L(S(t), I(t), \sigma(t))$ is the instantaneous cost at time $t$, and the extra term $S_\infty(S(T), I(T), \sigma_0)$ incentivizes the state of the system at time $T$ to lead to a favorable long-term trajectory (in the absence of any interventions after time $T$). In [27], the following formula for $S_\infty$ is given; see that paper for further discussion:

$$S_\infty(S, I, \sigma_0) = \frac{W_0(-\sigma_0 I e^{-\sigma_0(S+I)})}{\sigma_0}.$$

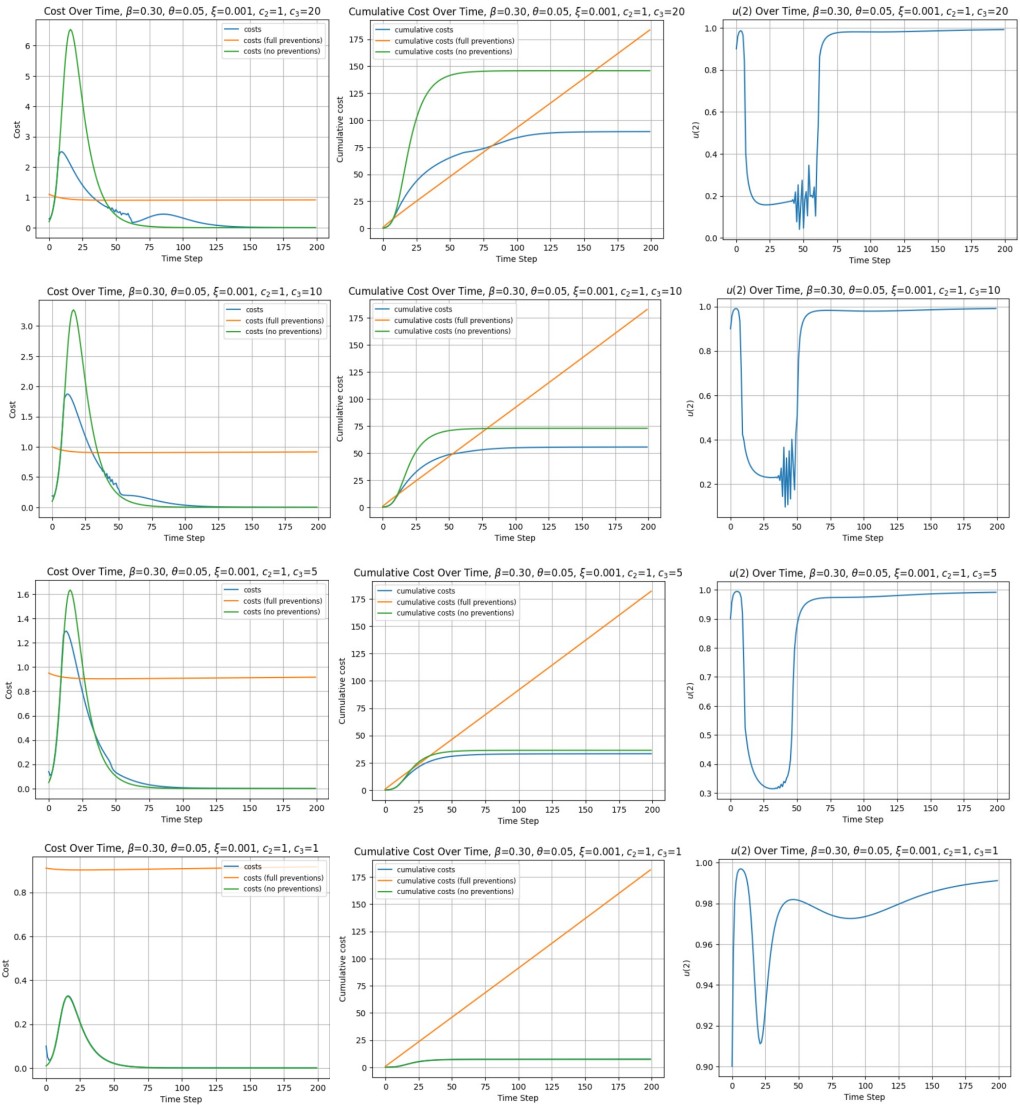

**Figure 6:** Control with costs: control over $T = 200$ steps. $\gamma_t = 0, \forall t$. SIR parameters: $\beta = 0.3, \theta = 0.05, \xi = 0.001$. Initial state $x_1 = [0.9, 0.1, 0]$. `GPC-Simplex` parameters: $H = 5$. **Left**: instantaneous cost over time, compared with that of no control (green) and full control (orange). **Middle**: cumulative cost over time. **Right**: $u_t(2)$ output by `GPC-Simplex` over time. $(c_2, c_3)$ values (from top to bottom rows): $(1, 20), (1, 10), (1, 5), (1, 1)$.

The instantaneous cost is modeled by [27] as follows:

$$L(S(t), I(t), \sigma(t)) = c_2 \cdot \left(1 - \frac{\sigma(t)}{\sigma_0}\right)^2 + \frac{c_3 \cdot (I(t) - y_{\max})}{1 + e^{-100(I(t) - y_{\max})}},$$

where $c_2, c_3$ are some parameters determining the cost of preventing disease transmission and the cost of a medical surge (i.e. when the proportion of infected individuals exceeds $y_{\max}$). Notice that the second term above will indeed be very small in magnitude unless $I(t)$ exceeds $y_{\max}$.

Note that `GPC-Simplex` cannot directly handle end-of-trajectory losses such as the term $S_\infty(S(T), I(T), \sigma_0)$. Thus, in our evaluation of `GPC-Simplex` on this system, we instead incorporate $S_\infty$ into the instantaneous cost functions. Concretely, we use the following cost function at

time $t$:

$$c_t(x_t, u_t) = -S_\infty(x_t(1), x_t(2), \sigma_0) + c_2 \cdot u_t(1)^2 + \frac{c_3(x_t(2) - y_{\max})}{1 + e^{-100(x_t(2) - y_{\max})}}.$$

Recall that we write $x_t = (S_t, I_t, R_t)$ and $u_t(1) = 1 - \sigma(t)/\sigma_0$, so modulo the addition of $S_\infty$ to all times $t < T$ and the conversion from continuous time to discrete time, our loss is analogous to that of [27].

# H  Experiments: Controlled replicator dynamics

The *replicator equation* is a basic model in evolutionary game theory that describes how individuals in a population will update their strategies over time based on their payoffs from repeatedly playing a game with random opponents from the population [11]. The basic principle is that strategies (or traits) that perform better than average in a given environment will, over time, increase in frequency within the population, whereas strategies that perform worse than average will become less common.

Formally, consider a normal-form two-player game with $d$ possible strategies and payoff matrix $M \in \mathbb{R}^{d \times d}$. A population at time $t$ is modelled by the proportion of individuals that currently favor each strategy, and thus can be summarized by a distribution $x(t) \in \mathbb{R}^d$. The *fitness* of an individual playing strategy $i \in [d]$ in a population with strategy distribution $x \in \Delta^d$ is defined to be

$$\text{fitness}_{M,x}(i) := e_i^\top M x,$$

where $e_i \in \mathbb{R}^d$ is the indicator vector for strategy $i$. That is, $\text{fitness}_{M,x}(i)$ is simply the expected payoff of playing strategy $i$ against a random individual from the population. The replicator dynamics posit that the population's distribution over strategies $x(t)$ will evolve according to the following differential equation:

$$\frac{dx_i(t)}{dt} := x_i(t) \cdot \left(\text{fitness}_{M,x(t)}(i) - \mathbb{E}_{j \sim x}\text{fitness}_{M,x(t)}(j)\right) = x_i(t) \cdot (e_i^\top M x(t) - x(t)^\top M x(t)). \tag{49}$$

It is straightforward to check that this differential equation preserves the invariant that $x(t)$ is a distribution. This equation can induce various types of dynamics depending on the initialization and payoff matrix $M$: the distribution may converge to an equilibrium, or it may cycle, or it may even exhibit chaotic behavior [11]. In this study we focus on a simple (time-discretized) replicator equation – namely, the equation induced by a generalized Rock-Paper-Scissors game – when the payoffs may be *controlled*.

**Controlled Rock-Paper-Scissors.**  The standard Rock-Paper-Scissors game has $d = 3$ and payoff matrix

$$M := \begin{bmatrix} 0 & 1 & -1 \\ -1 & 0 & 1 \\ 1 & -1 & 0 \end{bmatrix}.$$

Consider a setting where the game is run by an external agent that is allowed to set the payoffs. For simplicity, we assume that the game remains zero-sum and the rewards sum to 1, so the payoff matrix is now

$$M(u) := \begin{bmatrix} 0 & u_1 & -u_3 \\ -u_1 & 0 & u_2 \\ u_3 & -u_2 & 0 \end{bmatrix}$$

for a control vector $u \in \Delta^3$. The discrete-time analogue of the replicator equation with this controlled payoff matrix $M(u)$ is

$$x_{t+1} = f(x_t, u_t) := x_t + \eta \begin{bmatrix} x_{t1} \cdot e_1^\top M(u_t)x_t \\ x_{t2} \cdot e_2^\top M(u_t)x_t \\ x_{t3} \cdot e_3^\top M(u_t)x_t \end{bmatrix} \tag{50}$$

where $x_t, u_t \in \Delta^3$ are the population distribution and control at time $t$ respectively, and $\eta \in (0, 1)$ is the rate of evolution. Note that the term $x_t M(u_t) x_t$ does not need to appear in Eq. (50) because $M(u_t)$ is always zero-sum. Also, since $\eta \leqslant 1$ and all entries of $M(u_t)$ are at most 1 in magnitude, if $x_t$ is a distribution then $x_{t+1}$ will remain a distribution. We omit noise in this study, so Eq. (50) is a special case of Eq. (9) with $\gamma_t = 0$ for all $t$.

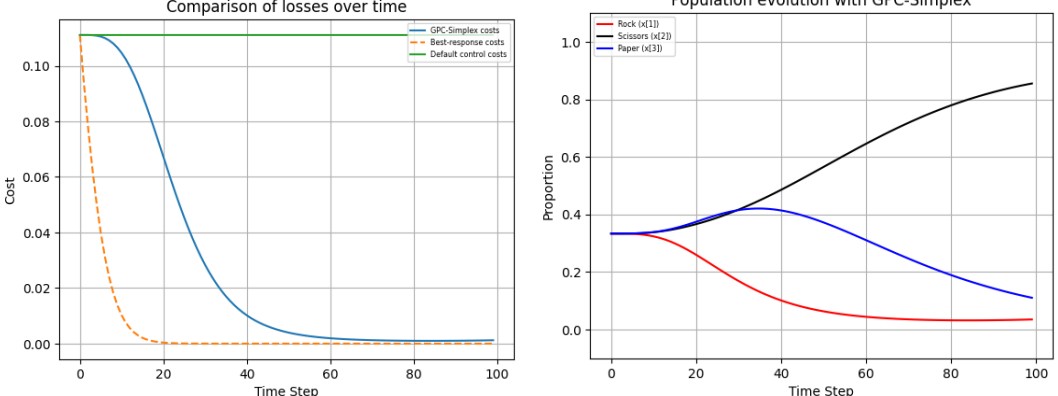

**(a)** Instantaneous cost achieved by `GPC-Simplex` over time, compared to default Rock-Paper-Scissors control and Best Response control (dashed orange).

**(b)** Proportions of the population playing strategies "rock", "paper", and "scissors" over time under control by `GPC-Simplex`.

**Figure 7:** Experimental results for dynamical system with horizon $T = 100$, uniform initial state, update rule Eq. (50) with $\eta = 1/4$, no perturbations, and time-invariant cost function $c_t(x_t, u_t) = x_{t1}^2$ for all times $t$. `GPC-Simplex` was implemented as described in Appendix E. The Best Response controller at each time $t$ picks the control $u$ that minimizes $c_{t-1}(f(x_t, u), u)$. The default controller picks the uniform control $u = (1/3, 1/3, 1/3)$.

**Parameters and cost function.** We define a (nonlinear) dynamical system with uniform initial state $x_1 = (1/3, 1/3, 1/3)$, update rule Eq. (50) with $\eta = 1/4$, and $T = 100$ timesteps. We consider the fixed cost function

$$c(x_t, u_t) := x_{t1}^2,$$

which can be thought of as penalizing the strategy "rock".

**Results.** We compare `GPC-Simplex` (implemented as described in Appendix E) with a baseline control that simply uses the standard Rock-Paper-Scissors payoff matrix (up to scaling) induced by $u = (1/3, 1/3, 1/3) \in \Delta^3$. As shown in Fig. 7a, `GPC-Simplex` (shown in blue) significantly outperforms this baseline (shown in green), learning to alter the payoff in such a way that the population tends to avoid the "rock" strategy. The evolution of the dsitribution over time under `GPC-Simplex` is shown in Fig. 7b.

For completeness, we also compare `GPC-Simplex` against the "Best Response" strategy (shown in dashed orange) that essentially performs 1-step optimal control, using the fact that the cost function for this example is time-invariant. While both controllers eventually learn a good policy, Fig. 7a clearly shows that Best Response learns faster. However, it is strongly exploiting the time-invariance of the cost function, since in general, this algorithm computes the best response with respect to the *previous* cost function rather than the *current* cost function, which it does not observe until after playing a control. In Fig. 8, we consider a slightly modified system where the cost function includes a cost on the control with probability $1/2$. In this setting, we see that `GPC-Simplex` still eventually learns a good policy, whereas Best Response and the default control incur large costs through the trajectory. Best Response in particular suffers greatly due to the time-varying nature of the costs.

# I  Discussions

## I.1  Broader impacts

Our work provides a robust algorithm with theoretical justifications for practical control problems that might be applicable to problems such as disease control. The experiments performed are preliminary. More careful empirical verification is necessary before our algorithm can be responsibly implemented in high-impact scenarios. Excluding the scenario of ill intention, we do not anticipate any negative social impact.

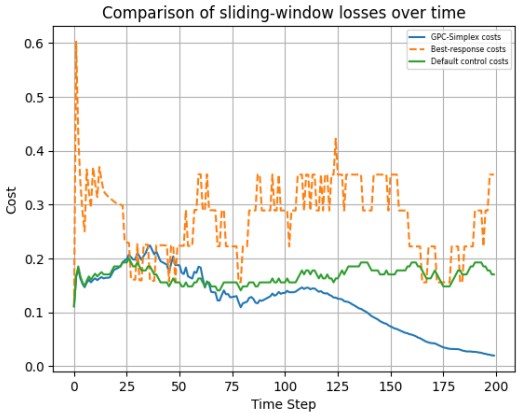

**Figure 8:** Experimental results for dynamical system with horizon $T = 200$, uniform initial state, update rule Eq. (50) with $\eta = 1/4$, no perturbations, and random cost function which is either $c_t(x_t, u_t) = x_{t1}^2$ or $c_t(x_t, u_t) = x_{t1}^2 + u_{t3}^2$ with equal probability. `GPC-Simplex` was implemented as described in Appendix E. The Best Response controller at each time $t$ picks the control $u$ that minimizes $c_{t-1}(f(x_t, u), u)$. The default controller picks the uniform control $u = (1/3, 1/3, 1/3)$. The plot shows the cost achieved by `GPC-Simplex` over time, compared to default Rock-Paper-Scissors control and Best Response control (dashed orange). Due to the non-continuity induced by the random cost functions, the loss plotted at time $t$ is the average loss of the controller across the last $\min(t, 15)$ time steps.

## I.2 Computational Resources for Experiments

The experiments in this work are simulations and relatively small-scaled. They were run on Google Colab with default compute resources. For each experiment, the time required to roll-out one trajectory using `GPC-Simplex` was less than 10 minutes.

