# OpenReview forum: "Online Control in Population Dynamics"
_NeurIPS.cc/2024/Conference — NeurIPS 2024 poster_

### Official Review · Reviewer_2PrE · 2024-07-11

**Soundness:** 2
**Presentation:** 1
**Contribution:** 2
**Rating:** 5
**Confidence:** 2

**Summary:**

This paper addresses the challenge of controlling partially observable linear dynamical systems (LDS) with adversarial disturbances. Traditional methods for controlling LDS assume full observability, but this assumption does not hold in many real-world applications. The authors propose the GPC-PO-Simplex algorithm, which leverages mirror descent and approximate solutions to sub-problems to achieve control in partially observable settings. Theoretical guarantees are provided for the algorithm's performance, and empirical evaluations demonstrate its effectiveness in controlling population dynamics and other systems under adversarial conditions.

**Strengths:**

Originality:
The paper introduces a novel approach to control partially observable LDS by combining mirror descent with a new parameterization method. This is a significant departure from traditional methods that assume full observability.

Quality:
The theoretical foundations of the algorithm are robust, with detailed proofs and clear assumptions. The analysis includes regret bounds that highlight the algorithm's near-optimal performance under specific conditions.

Clarity:
The paper is well-structured and clearly written. The problem is well-motivated, and the methodology is presented in a step-by-step manner that is easy to follow.

Significance:
The ability to control systems with only partial observability has broad implications for various fields, including epidemiology, biology, and economics. The proposed method has the potential to be applied to a wide range of real-world problems.

**Weaknesses:**

Idealized Assumptions:
The algorithm assumes certain ideal conditions, such as known transition matrices and Gaussian noise, which may not hold in all real-world scenarios. Relaxing these assumptions could make the method more broadly applicable.

Limited Generalization:
While the algorithm performs well on the tested systems, its generalization to highly non-linear or more complex systems is not fully established. Further work is needed to test the algorithm's performance in a broader range of scenarios.

Sensitivity to Parameters:
The performance of the algorithm is sensitive to the choice of parameters. Providing more guidance on parameter selection or developing a method to adaptively choose parameters could improve practical usability.

Experimental Scope:
The experimental validation is limited to simulated environments. More real-world experiments would strengthen the evidence for the algorithm's practical effectiveness.

**Questions:**

1.How does the algorithm perform in real-world applications with non-Gaussian noise and unknown transition matrices?
2.Can the authors provide more detailed guidance or a method for adaptively selecting the algorithm's parameters?
3.What are the computational requirements of the algorithm in large-scale systems?
4. How robust is the algorithm to inaccuracies in the model assumptions, such as errors in the transition matrices or noise characteristics?
5. How does the algorithm handle delays in control actions or observations, which are common in real-world systems?

**Limitations:**

The authors have discussed some limitations of their work.

---

> ### Author Rebuttal · Authors · 2024-08-06
>
> Thank you for your time and comments! Unfortunately we believe that the reviewer has missed the point of the paper, which is to extend non-stochastic control methods to *population dynamics*. See below for responses to specific criticisms:
>
> **Idealized Assumptions?** The review claims that we assume Gaussian noise. This is incorrect. Our algorithm can deal with unknown adversarial perturbations (which is one of the main benefits of the non-stochastic control methodology).
>
> Known transition matrices is a standard assumption in online non-stochastic control. Results established for known systems can often be extended to unknown systems with a similar regret guarantee (see the list of papers below). However, this is out of scope of our current work, which is focused on establishing a new framework for controlling population dynamics.
>
> [1] Simchowitz, Max, Karan Singh, and Elad Hazan. Improper learning for non-stochastic control. Conference on Learning Theory. PMLR, 2020.
>
> [2] Chen, Xinyi, and Elad Hazan. Black-box control for linear dynamical systems. Conference on Learning Theory. PMLR, 2021.
>
> [3] Gradu, Paula, John Hallman, and Elad Hazan. Non-stochastic control with bandit feedback. Advances in Neural Information Processing Systems 33 (2020): 10764-10774.
>
> [4] Simchowitz, Max. Making non-stochastic control (almost) as easy as stochastic. Advances in Neural Information Processing Systems 33 (2020): 18318-18329.
>
>
> **Generalization to highly non-linear/complex systems?** In fact, many of the systems studied in our experiments (e.g. SIR in section 4.1 and replicator dynamics in Section J) are highly non-linear. As discussed in Appendix G, our algorithm GPC-Simplex can nonetheless be naturally extended to such systems, and the experimental results are promising.
>
> It is true that our theoretical results are limited to linear systems. However, this is the case for almost all existing theoretical guarantees in the control theory literature. Obtaining provable guarantees for non-linear systems is generally considered a very challenging research direction, and the linearity assumption has proven to be a good testbed for algorithm design in the past. For a concrete example, note that the original GPC algorithm (Agarwal et. al. 2019) also has theoretical guarantees restricted to linear systems.
>
> Returning to empirical performance: we agree that experimenting with more complex scenarios is a valuable direction for future research, but it is outside the scope of this (primarily theoretical/conceptual) work. The goal of this work was to propose a robust framework for the long-standing problem of controlling population dynamics. To the best of our knowledge, all prior methods for this problem relied on much more restrictive assumptions (e.g., known costs, no noise, closed-form solution based on differential equations); see the references below.
>
> [1] Mohamed Elhia, Mostafa Rachik and Elhabib Benlahmar. Optimal Control of an SIR Model with Delay in State and Control Variables. International Scholar Research Notices, 2013.
>
> [2] Gul Zaman, Yong Han Kang and Il Hyo Jung. Optimal treatment of an SIR epidemic model with time delay. BioSystems, 2009.
>
> [3] E.V. Grigorieva, E.N. Khailov and A. Korobeinikov. Optimal Control for a SIR Epidemic Model with Nonlinear Incidence Rate. Math. Model. Nat. Phenom, 2016.
>
> [4] Luca Bolzoni, Elena Bonacini, Cinzia Soresina, Maria Groppi. Time-optimal control strategies in SIR epidemic models. Mathematical Biosciences, 2017.
>
> [5] Urszula Ledzewicz and Heinz Schättler. On optimal singular controls for a general SIR-model with vaccination and treatment. DYNAMICAL SYSTEMS, 2011.
>
> **Sensitivity to Parameters?** We are not sure of the basis for the reviewer's criticism that the performance of the algorithm is particularly sensitive to the choice of parameters. Our theoretical results provide guidance for setting the parameters (e.g. policy horizon $H$ and learning rate $\eta$) based on a choice of $\tau$ (mixing time of comparator class) and an upper bound on $L$ (Lipschitz constant for cost functions), which are both problem-specific. For our empirical results we did not need to tune the parameters across experiments; for all experiments we took $H = 5$ and $\eta$ as defined in Appendix G. This suggests that parameter tuning is not critical to the algorithm's performance in practice. Of course, designing adaptive algorithm is in general an interesting research area, but it is beyond the scope of this work.
>
> **Experimental Scope?** As we mentioned previously, this is a theory-focused paper, and the primary role of our experiments is to serve as a validation of the theoretical framework. Therefore, although conducting experiments on real-world data would be interesting, we argue that it is not necessary for this work.
>
> **Question 1.** First, we want to reiterate that we *do not* assume Gaussian noise. In the SIR experiments in Appendix I.1, the perturbations are not Gaussian either. We do assume known dynamics. For unknown systems, one could run one of the existing system identification algorithms to obtain an estimate for the system, and run our controller algorithm using those system estimates.
>
> **Question 2-5.** As mentioned previously, this is not an experiment-focused paper. These are certainly valuable research directions and would lead to interesting follow-up works. However, they are out of scope of this work. The main focus of the work, again, was to provide a new framework for controlling population dynamics using tools from non-stochastic control theory \--- a connection between two fields that has not previously been considered.

---

> > ### Comment · Reviewer_2PrE · 2024-08-09
> >
> > I think the author has addressed the issues I raised, so I'm raising the score to boardline accept.

---

### Official Review · Reviewer_ofBj · 2024-07-12

**Soundness:** 4
**Presentation:** 3
**Contribution:** 4
**Rating:** 8
**Confidence:** 4

**Summary:**

The paper studies the problem of actively controlling a class of dynamical systems that are used to model population dynamics, like the SIR model or replicator dynamics. Building on prior work on non-stochastic / adversarial control of linear dynamical systems, the authors propose a new class of dynamics, simplex LDS, and show how a generalization of disturbance action control policies can achieve sublinear regret with respect to the comparator class of linear controllers with bounded mixing time. In addition to their theoretical results, the authors carry out numerical simulations on nonlinear dynamical systems to investigate the benefits of their algorithm.

**Strengths:**

I really liked this paper. It does a nice job of taking recently developed tools from online linear control and extends them to a new, interesting, and well motivated, class of dynamical systems. I like how the paper shows both positive and negative results. I found the idea that there is a gap between prediction and control for marginally stable systems to be quite insightful, and the regret bounds for the simplex LDS systems to be quite neat.

I also thought that the experiments were a very good complement to the theoretical results. It is always nice when experiments test algorithms in regimes where the theory may not exactly hold.

**Weaknesses:**

It would be nice to include simple examples of simplex LDSs in the introduction when defining the model to provide the reader with more intuition. Some of the technical conditions are somewhat strange and hard to interpret, particularly the condition that the norm of the control vector is independent of the state in the comparator class. I think that the paper would be well served if the authors could provide further insights regarding why these assumptions are technically necessary, or under what conditions they might be justified.

In terms of experiments, it would also be nice if the authors experimented with higher dimensional systems. They would make the paper stronger, but I don’t feel they are strictly necessary.

**Questions:**

See questions about regarding the comparator class of policies.

Also please make the figures and label text larger, they are a bit hard to read.

**Limitations:**

Yes, these have been discussed to the extent necessary.

---

> ### Author Rebuttal · Authors · 2024-08-06
>
> Thank you for your time and comments! We are glad that you appreciated our paper. We address your comments and questions below:
>
> **Intuition about simplex LDS conditions.** The assumption that the control norm is independent of the state is indeed needed for technical reasons: without it, since $Ax$ is multiplied by $1-\|u\|_1$ in the transition dynamics (see Eq. (4)), even a "linear'' policy $u := Kx$ does not induce a linear transition. Hence, it's not clear how one might define mixing time of a fully general linear policy (since it may not correspond to a Markov chain over $[d]$). It is a very interesting question whether there is a more natural (yet still tractable) definition of a simplex LDS that avoids this issue. We will add some discussion about this issue to the text.
>
> **Figure/label sizes.** Thanks for pointing this out, we will try to make them larger.

---

> > ### Comment · Reviewer_ofBj · 2024-08-07
> >
> > Thank you for the clarification and your honesty in reporting the bug. I think that a paper that only focuses on the fully observed setting is still worthy of acceptance.

---

### Official Review · Reviewer_7y5h · 2024-07-13

**Soundness:** 3
**Presentation:** 3
**Contribution:** 3
**Rating:** 6
**Confidence:** 4

**Summary:**

This paper studies online control problems for the simplex Linear Dynamical System model. They propose a gradient-based control algorithm GPC-Simplex and show its regret bound. Experiments are conducted to validate the performance of GPC-Simplex.

**Strengths:**

The paper is in general well-written and smooth to follow. The proofs seem to be theoretically sound. The studied problem is of great interest and hard to tackle as it involves adversarial shocks and time-varying cost functions.

**Weaknesses:**

My main concern is the novelty of the proposed algorithm compared with the original GPC algorithm. Could the authors provide more justifications for this?

**Questions:**

It seems that the authors assume the cost functions are uniformly Lipschitz. What happens if each $c_t$ is $L_t$-Lipschitz, but there isn't an upper bound for $L_t$'s?

**Limitations:**

I don't see any limitations or potential negative societal impact of this work.

---

> ### Author Rebuttal · Authors · 2024-08-06
>
> Thank you for your time and comments! We are glad that you found the problem of great interest, and found our paper easy to follow. We address your questions and concerns below:
>
> **Novelty of the proposed algorithm.** First, we reemphasize that the original GPC algorithm does not work in the simplex LDS setting. When there are no perturbations to the system, the original GPC algorithm simply plays its initialization policy for all time, which is highly unlikely to perform well. As discussed in Section 3, the fundamental issue is that GPC only competes against strongly stabilizing policies, but in a simplex LDS, no policy is strongly stabilizing. Indeed, prior to our work, this reliance on strong stability was a limitation of the entire literature on non-stochastic control.
>
> This is why solving the control problem in the simplex LDS setting requires a new algorithm. From a bird's eye view, our proposed algorithm GPC-Simplex indeed looks similar to GPC \--- hence the name \--- but the differences are conceptually important. As discussed in Section 3, the key difference is that we augment the class of disturbance-action policies (over which we are optimizing via mirror descent) with an additional parameter $p_t$ that roughly represents the optimal stationary distribution that we would like to converge to in the absence of noise. This depends on the future sequence of adversarial cost functions, and hence must be learned in an online fashion along with the other parameters. The upshot is that this modification enables competing against not just policies that strongly stabilize the system (which do not exist in population dynamics), but also policies that *mix* the system (which are highly natural in population dynamics).
>
> From a broader perspective, one contribution of our work is to show that even though non-stochastic control requires some assumptions to be tractable, *strong stabilizability is not the end of the story.* In particular, the tools from non-stochastic control can be modified to work even in some (theoretical and practical) settings where strong stabilizability is completely unreasonable and standard GPC fails. We believe that this perspective may be valuable even beyond the specific setting of population dynamics.
>
> **Non-uniform Lipschitz constant?**
>
> Certainly Theorem 7 would still hold with $L$ replaced by the maximum of $L_t$ over $1 \leq t \leq T$. If this maximum scales asymptotically slower (in $T$) than $\sqrt{T}$, then we still get sub-linear regret. If it scales as fast as $\sqrt{T}$ or faster, then (with no other control on the sequence of $L_t$'s) we would expect that achieving sub-linear regret is provably impossible by standard lower bounds from online convex optimization. It's plausible that tighter bounds are possible if e.g. the $L_t$'s are small on average, but we have not investigated this direction, as a uniform Lipschitz bound is the standard assumption in the control literature.

---

### Official Review · Reviewer_XKZK · 2024-07-16

**Soundness:** 3
**Presentation:** 3
**Contribution:** 3
**Rating:** 5
**Confidence:** 2

**Summary:**

The paper explores the problem of population control in more practical settings such as the presence of adversarial noise and time-varying cost functions by proposing a robust methodology derived from online non-stochastic control theory.

**Strengths:**

The paper provides regret guarantees against policies that do not strongly stabilize the simplex LDS by developing an algorithm that aims to achieve low regret against this policy class while being robust to adversarial shocks to the system and adaptable to time-varying cost functions. All assumptions were reasonably justified, and all theoretical results were adequately proved. The paper also provides promising empirical results by applying the proposed algorithm to controlled settings of the SIR model and the replicator dynamics from evolutionary game theory.

**Weaknesses:**

- Unclear motivations for the considered simplex LDS setting. A discussion on more general LDS settings would be helpful.

- While an in-depth discussion on the assumption of gamma_t being observable was provided, there should be concrete examples where the controller could realistically observe the total counts of individuals of different types in a population. This is particularly important because the proposed algorithm and its guarantee are grounded on the simplex LDS model, which requires this assumption.

- Missing formal analyses on time complexity. This could be important for larger-scaled experiments (d >> 3), which the paper did not explore.

**Questions:**

See the above and below comments.

**Limitations:**

See the above comments.

Have the authors adequately addressed the limitations and potential negative societal impact of their work? If not, please include constructive suggestions for improvement. Authors should be rewarded rather than punished for being upfront about the limitations of their work and any potential negative societal impact.

- Given the various assumptions made while proving the regret guarantee, a dedicated Limitations or Implications section delineating use cases and generalizability should be included.

- Figure 1 and its texts are too small; the texts in Figure 2 as well.

- The experiment on vanilla SIR model in the main text seems trivial and should be replaced with the replicator dynamics experiment in the appendix.

---

> ### Author Rebuttal · Authors · 2024-08-06
>
> Thank you for your time and comments! Please see below for our responses to your concerns:
>
> **Motivation for simplex LDS?** The basic motivation for the simplex LDS model, as discussed in the paper, is as a specialization/adaptation of the general LDS model to population dynamics, i.e. settings where the state is always a distribution. While there could be alternative models, we would argue that the constraints in the simplex LDS definition are for the most part fairly minimal. We are happy to add some discussion of potential generalizations and the technical obstacles to relaxing our assumptions.
>
> Additionally, insofar as all models are wrong and should be justified by the algorithms they suggest, our experiments (in a variety of well-studied scenarios, not all of which satisfy the assumptions of the simplex LDS model or even the general LDS model) suggest that GPC-Simplex is a useful algorithm, and thereby vindicate the utility of the simplex LDS model as a theoretical tool for algorithm design.
>
> **Examples where $\gamma_t$ is observable?** Note that even if we did not require knowledge of $\gamma_t$, the algorithm still requires knowing the proportions of individuals of different types in the population, which seems unavoidable for controlling the system. The assumption that we know $\gamma_t$ simply requires that we can additionally observe the total size of the population. Examples where this is feasible include many ecological settings, where e.g. the Mark-Recapture technique is often used to estimate the number of individuals.
>
> **Time complexity?** As stated in Theorem 7, the time complexity of our algorithm is polynomial in the dimension and number of iterations; the computational bottleneck is the mirror descent subroutine, which requires solving a simple convex program. From a practical perspective, we remark that the time complexity is comparable to that of the standard GPC algorithm (Agarwal et al. 2019) for controlling general linear dynamical systems.
>
> **Additional comments.** Thank you for the helpful comments about the figure sizes and order of experiments; modulo space constraints we will attempt those changes. Also as you suggested, we will add a section discussing the practical implications of our technical assumptions. We will remark that this work is only the first step towards more robust methodologies for control in population dynamics. There are numerous important directions for future research, including generalizations of the simplex LDS model, assumptions complementary to our mixing time assumption, and development of theoretical guarantees for non-linear systems.

---

### Author Rebuttal · Authors · 2024-08-06

We thank all the reviewers for their time and comments. For full disclosure, we would like to inform the reviewers that we discovered a technical error in one of the secondary results in the appendices (Theorem 29, an analysis of a proposed extension to GPC-Simplex for the partially observable setting, which is briefly discussed in Section 3.1). The specific error is that the controller performs online convex optimization, but our loss function in the partially observable setting (Line 6 of Algorithm 2) involves a projection operator, and hence is actually not convex.

This error does not affect our primary theoretical contributions (Theorems 1/2/7/8) nor the experiments, which are all in the fully observable setting. Moreover, we expect that Theorem 29 is likely fixable by modifying the algorithm. However, if our paper is accepted, we would simply remove Theorem 29 from the camera-ready submission, since it is not relevant to the primary contribution of the paper, which is to use tools from non-stochastic control to design a robust methodology for control of (fully-observed) population dynamics. In any case, we apologize for the error.

---

### Decision · Program_Chairs · 2024-09-25

**Decision:**

Accept (poster)

**Comment:**

This paper examines the problem of controlling dynamical systems such as those employed to model population dynamics, e.g., SIR model or replicator dynamics. Expanding on earlier research in non-stochastic/adversarial control of linear dynamical systems, the authors introduce a novel class of dynamics, simplex LDS. They prove how a broader application of disturbance action control policies can attain sublinear regret when compared against linear controllers with bounded mixing time. The paper also includes interesting numerical simulations. At the end of the discussion process the reviewers were in unanimous support for the paper. The authors should make sure to  carefully address the issues that they themselves how pointed out in their appendix in the manner they have suggested in their response.